

# Underestimation of brown carbon absorption based on the
methanol extraction method and its impacts on source analysis
Zhenqi Xu[a], Wei Feng[a], Yicheng Wang[a], Haoran Ye[a], Yuhang Wang[b], Hong Liao[a],
Mingjie Xie[a,*]
[a]Collaborative Innovation Center of Atmospheric Environment and Equipment
Technology, Jiangsu Key Laboratory of Atmospheric Environment Monitoring and
Pollution Control, School of Environmental Science and Engineering, Nanjing
University of Information Science & Technology, 219 Ningliu Road, Nanjing 210044,
China
[b]School of Earth and Atmospheric Sciences, Georgia Institute of Technology, Atlanta,
GA 30332, United States
*Corresponding to:
Mingjie Xie (mingjie.xie@nuist.edu.cn, mingjie.xie@colorado.edu);
Mailing address: 219 Ningliu Road, Nanjing, Jiangsu, 210044, China


**Abstract**

The methanol extraction method was widely applied to isolate organic carbon (OC) from ambient aerosols, followed by measurements of brown carbon (BrC) absorption. However, undissolved OC fractions will lead to underestimated BrC absorption. In this work, water, methanol (MeOH), MeOH/dichloromethane (MeOH/DCM, 1:1, v/v), MeOH/DCM (1:2, v/v), tetrahydrofuran (THF), and N,N-dimethylformamide (DMF) were tested for extraction efficiencies of ambient OC, and the light absorption of individual solvent extracts was determined. Among the five solvents and solvent mixtures, DMF dissolved the highest fractions of ambient OC (up to ~95%), followed by MeOH and MeOH/DCM mixtures (< 90%), and the DMF extracts had significant ($p < 0.05$) higher light absorption than other solvent extracts. This is because the OC fractions evaporating at higher temperatures (> 280℃) are less soluble in MeOH (~80%) than in DMF (~90%) and contain stronger light-absorbing chromophores. Moreover, the light absorption of DMF and MeOH extracts of collocated aerosol samples in Nanjing showed distinct time series. Source apportionment results indicated that the MeOH insoluble OC mainly came from unburned fossil fuels and polymerization processes of aerosol organics. These results highlight the necessity of replacing MeOH with DMF for further investigations on structures and light absorption of low-volatile BrC.



## 1 Introduction

Besides black carbon (BC) and mineral dust, growing evidence shows that organic carbon (OC) aerosols derived from various combustion sources (e.g., biofuel and fossil fuel) and secondary processes (e.g., gas-phase oxidation, aqueous and in-cloud processes) can absorb sunlight at short visible and UV wavelengths (Laskin et al., 2015; Hems et al., 2021). The radiative forcing (RF) of the light-absorbing organic carbon, also termed "brown carbon" (BrC), is not well quantified due to the lack of its emission data and large uncertainties in *in situ* BrC measurements (Wang et al., 2014; Wang et al., 2018; Saleh, 2020). The imaginary part of the refractive index ($k$) of BrC is required when modeling its influence on aerosols direct RF, and is retrieved by the optical closure method combing online monitoring of aerosol absorption and size distributions with Mie theory calculations (Lack et al., 2012; Saleh et al., 2013; Saleh et al., 2014). However, several pre-assumptions must be made on aerosol morphology (spherical Mie model) and mixing states of BC and organic aerosols (OA), which might introduce large uncertainties in the estimation of $k$ (Mack et al., 2010; Xu et al., 2021).

To improve the understanding on chemical composition and light-absorbing properties of BrC chromophores, organic matter (OM) in aerosols was isolated through solvent extraction using water and/or methanol, followed by filtration and a series of instrumental analysis (e.g., UV/Vis spectrometer, liquid chromatograph-mass spectrometer; Chen and Bond, 2010; Liu et al., 2013; Lin et al., 2016). Referring to existing studies, a larger fraction of the methanol extract absorption comes from water-insoluble OM containing conjugated structures (Chen and Bond, 2010; Huang et al., 2020); the light absorption of biomass burning OM is majorly contributed by large molecules (MW > 500~1000 Da; Di Lorenzo and Young, 2016; Di Lorenzo et al., 2017) and depends on burn conditions (Saleh et al., 2014); polycyclic aromatic hydrocarbons





(PAHs) and nitroaromatic compounds (NACs) are ubiquitous BrC chromophores in the
atmosphere (Huang et al., 2018; Wang et al., 2019), but the identified species only
explain a few percentages (< 10%) of total BrC absorption (Huang et al., 2018; Li et
al., 2020).

Methanol can extract > 90% OM from biomass burning (Chen and Bond, 2010;

Xie et al., 2017b), while the extraction efficiency ($\eta$, %) decreases to ~80% for ambient
organic aerosols (Xie et al., 2019b; Xie et al., 2022) possibly due to other sources
emitting large hydrophobic molecules and oligomerizations of small molecules during
the aging process (Cheng et al., 2021; Li et al., 2021). The light-absorbing properties
and structures of methanol-insoluble OC (MIOC) are still unknown. By comparing BrC
characterization results of offline and online methods, some studies conclude that the
MIOC dominates BrC absorption in source and ambient aerosols (Bai et al., 2020; Atwi
et al., 2022). However, the online-retrieval and offline-extraction methods are designed
based on different instrumentation and purposes, and the online method depends largely
on presumed and uncertain optical properties of BC (Wang et al., 2014). Thus, BrC
absorption in particles and solution can hardly be compared directly. To reveal the
absorption and composition of MIOC, it is necessary to find a new solvent or develop
a new methodology to improve OC extraction efficiency (Shetty et al., 2019).

In this work, a series of single solvents and solvent blends were tested for extraction

efficiencies of OC in ambient particulate matter with aerodynamic diameter < 2.5 μm
($PM_{2.5}$), and the sample extract absorption of each solvent was compared. The solvent
or solvent mixture with the highest $\eta$ value was applied to extract a matrix of collocated
$PM_{2.5}$ samples followed by light absorption measurements. In our previous work, the
light absorption of methanol extracts of the same samples was measured, and source
apportionment was performed using organic molecular marker data (Xie et al., 2022).

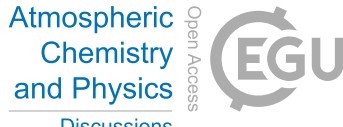
By comparing with the study results in Xie et al. (2022), this study evaluated the
underestimation of BrC absorption in methanol and its impacts on BrC source
attributions. These results suggest that methanol should be replaced in future solvent
extraction-based investigations on the absorption, composition, sources, and formation
pathways of low-volatile BrC.

**2. Methods**

*2.1 Solvent selection*

Five solvents and solvent mixtures including water, methanol (MeOH),
MeOH/dichloromethane (MeOH/DCM, 1:1, v:v), MeOH/DCM (1:2, v:v),
tetrahydrofuran (THF), and N,N-dimethylformamide (DMF) were selected to extract
OC from identical PM$_{2.5}$ samples to determine which solvent or solvent mixture has the
highest η value. Water and methanol are the most commonly used solvents to extract
BrC from source or ambient particles. Cheng et al. (2021) found that OC produced
through the combustion of toluene, isooctane, and cyclohexane were more soluble in
DCM than MeOH. Since a major part of BrC absorption is coming from unknown large
molecules (Di Lorenzo and Young, 2016; Di Lorenzo et al., 2017), polar aprotic
solvents THF and DMF were tested due to their high capacity for dissolving large
polymers. Except for water and MeOH, MeOH/DCM mixtures, THF, and DMF were
rarely used to extract OC for light absorption measurements.

*2.2 Sampling*

**Sampling for solvent test.** To compare OC extraction efficiencies and extract

absorption of the five selected solvents and solvent mixtures, twenty-one ambient PM$_{2.5}$
samples were collected on the rooftop of a seven-story library building in Nanjing
University of Information Science and Technology (NUIST, 32.21ºN, 118.71ºE).
Details of the sampling site and equipment were provided by Yang et al. (2021). Two


identical mid-volume samplers (Sampler I and II; $PM_{2.5}$-PUF-300, Mingye
Environmental, China) equipped with 2.5 μm cut-point impactors were used for
ambient air sampling during day-time (8:00 a.m.–7:00 p.m.) and night-time (8:00 p.m.–
7:00 a.m. the next day), respectively, in December 2019. After the impactor, $PM_{2.5}$ in
the air stream was collected on a pre-baked (550 °C, 4 h) quartz filter (20.3 cm ×12.6
cm, Munktell Filter AB, Sweden) at a flow rate of 300 L $min^{-1}$. $PM_{2.5}$ filter and field
blank samples were sealed and stored at –20 °C before chemical analysis. Information
about $PM_{2.5}$ samples for the solvent test is provided in Table S1 of supplementary
information.
**Ambient sampling for BrC analysis.** Details of the ambient sampling were described
in previous work (Qin et al., 2021; Yang et al., 2021; Xie et al., 2022). Briefly, Sampler
I and II were equipped with two quartz filters in series (quartz behind quartz, QBQ
method; $Q_f$ and $Q_b$) followed by adsorbents. Collocated filter and adsorbent samples
were collected every sixth day during daytime and nighttime from 2018/09/28 to
2019/09/28. Field blank sampling was performed every $10^{th}$ sample to address
contamination. $Q_f$ samples loaded with $PM_{2.5}$ were speciated and extracted for light
absorption measurements. The OC adsorbed on $Q_b$ and its light absorption were
analyzed to determine positive sampling artifacts. The adsorbents in sampler I [a
polyurethane foam (PUF)/XAD-4 resin/PUF sandwich] and II (a PUF plug) were used
to collect gas-phase nonpolar and polar organic compounds, respectively.
*2.3 Solvent test for light absorption and extraction efficiency*
An aliquot (~6 $cm^2$) of each filter sample was extracted ultrasonically in 10 mL of
each solvent or solvent mixture (HPLC grade) for 30 min (one-time extraction
procedure, $N = 11$; Table S1). After filtration, the light absorbance ($A_\lambda$) of individual
solvent extracts was measured over 200–900 nm using a UV/Vis spectrometer (UV-



1900, Shimadzu Corporation, Japan), and was converted to light absorption coefficient
($Abs_\lambda$, $Mm^{-1}$) by
$$Abs_\lambda = (A_\lambda - A_{700}) \times \frac{V_l}{V_a \times L} \ln(10) \qquad (1)$$
where $A_{700}$ is subtracted to correct baseline drift, $V_l$ ($m^3$) is the air volume of the
extracted sample, $L$ (0.01 m) is the optical path length, and ln (10) was multiplied to
transform $Abs_\lambda$ from a common to a natural logarithm (Hecobian et al., 2010). To
understand if multiple extractions could draw out more BrC, a two-time extraction
procedure was applied for another 10 ambient $PM_{2.5}$ samples in the same manner (Table
S1). The $A_\lambda$ of the 1st and 2nd extractions (10 mL each) was measured separately for
$Abs_\lambda$ calculations.
Prior to solvent extractions, the concentrations of OC and EC in each filter sample
were analyzed using a thermal-optical carbon analyzer (DRI, 2001A, Atmoslytic,
United States) following the IMPROVE-A protocol. OC and EC were converted to $CO_2$
step by step during two separate heating cycles [OC1 (140°C) – OC2 (280°C) – OC3
(480°C) – OC4 (580°C) in pure He, EC1 (580°C) – EC2 (740°C) – EC3 (840°C) in 98%
He/2% $O_2$], and the emitted $CO_2$ during each heating step was converted to $CH_4$ and
measured using a flame ionization detector (FID).
After extractions, filters extracted by MeOH, MeOH/DCM (1:1), MeOH/DCM
(1:2), and THF were air-dried in a fume hood and analyzed for residual OC (rOC, μg
$m^{-3}$) using the identical method. Filters extracted in water and DMF cannot be air-dried
in the short term due to the low volatility of solvents, and their rOC was measured after
baking at 100 °C for 2 h. The total amount of OC dissolved in water for each sample
was also measured as water-soluble OC (WSOC) by a total organic carbon analyzer
(TOC-L, Shimadzu, Japan; Yang et al., 2021). To examine if the baking process would



influence rOC measurements, the rOC of filters extracted in MeOH, MeOH/DCM
mixtures, and THF were also measured after the baking process and compared to those
determined after air dried. The pyrolytic carbon (PC) was used to correct for sample
charring and was determined when the filter transmittance or reflectance returned to its
initial value during the analysis (Schauer et al., 2003), but the formation of PC is very
scarce when analyzing extracted filters. In this study, solvent-extractable OC (SEOC,
$\mu$g m$^{-3}$) was determined by the difference in OC1–OC4 between pre- and post-
extraction samples. The extraction efficiency ($\eta$, %) of each solvent was expressed as
$$\eta = \frac{SEOC}{OC} \times 100\% \qquad (2)$$
Here, SEOC denotes WSOC when the solvent is water. For the ambient samples
extracted twice, rOC was measured only after the two-extraction procedure was
completed.

The solution mass absorption efficiency (MAE$_\lambda$, m$^2$ g$^{-1}$ C) was calculated by

dividing Abs$_\lambda$ by the concentration of SEOC
$$MAE_\lambda = \frac{Abs_\lambda}{SEOC} \qquad (3)$$
and the solution absorption Ångström exponent (Å), a parameter showing the
wavelength dependence of solvent extract absorption, was obtained from the regression
slope of lg (Abs$_\lambda$) versus lg ($\lambda$) over 300–550 nm.

To evaluate the influence of solvent effects on light absorption of different solvent

extracts of the same sample, solutions of 4-nitrophenol at 1.90 mg L$^{-1}$, 4-nitrocatechol
at 1.84 mg L$^{-1}$, and 25-PAH mixtures (Table S2) at 0.0080 mg L$^{-1}$ and 0.024 mg L$^{-1}$
(each species) in the five solvents and solvent mixtures were made up for five times
and analyzed for UV/Vis spectra. The absorbance of PAH mixtures in water was not
provided due to their low solubility.



*2.3 Measurements and analysis of ambient BrC absorption*

Collocated $Q_f$ and $Q_b$ samples were extracted using the solvent with the highest η value once followed by light absorbance measurement. OC concentrations in $Q_f$ and $Q_b$ samples were obtained from Yang et al. (2021), and SEOC values were estimated from OC concentrations and the average η value determined in *section 2.1* for one-time extraction. In this work, $Q_b$ measurements were used to correct $Abs_\lambda$, $MAE_\lambda$, and Å of BrC in ambient $PM_{2.5}$ in the same manner as those for water and methanol extracts in Xie et al. (2022)

Artifact-corrected $Abs_\lambda = Abs_\lambda^{Qf} - Abs_\lambda^{Qb}$ (4)

Artifact-corrected $MAE_\lambda = \dfrac{Abs_\lambda^{Qf} - Abs_\lambda^{Qb}}{SEOC_{Qf} - OC_{Qb}}$ (5)

where $Abs^{Qf}_\lambda$ and $Abs^{Qb}_\lambda$ are $Abs_\lambda$ values of $Q_f$ and $Q_b$ samples, respectively; $SEOC_{Qf}$ represents SEOC concentrations in $Q_f$ samples; $OC_{Qb}$ denotes OC concentrations in $Q_b$ samples, assuming that OC in $Q_b$ is completely dissolved (Xie et al., 2022). Artifact corrected Å were generated from the regression slope of lg ($Abs^{Qf}_\lambda$ - $Abs^{Qb}_\lambda$) versus lg (λ) over 300 – 550 nm. Artifact-corrected $Abs_\lambda$, $MAE_\lambda$, and Å during each sampling interval were determined by averaging each pair of collocated measurements. If one of the two numbers in a pair is missed, the other number will be directly used for the specific sampling interval. To compare with previous studies based on water and/or methanol extraction methods, $Abs_\lambda$ and $MAE_\lambda$ at 365 nm were shown and discussed in this work.

Pearson's correlation coefficient (*r*) was used to show how collocated measurements of BrC in ambient $PM_{2.5}$ vary together. The coefficient of divergence (COD) was calculated to indicate consistency between collocated measurements. The relative uncertainty of BrC absorption derived from duplicate data was depicted using


the average relative percent difference (ARPD, %), which was used as the uncertainty
fraction for BrC measurements. Calculation methods of COD and ARPD are provided
in Text S1 of supplementary information. To examine the influence of BrC
underestimation based on the methanol extraction method on source apportionment,
positive matrix factorization (PMF) version 5.0 (U.S. Environmental Protection
Agency) was applied to attribute the light absorption of aerosol extracts in methanol
and solvent with the highest η to sources. The input bulk components and organic
molecular marker (OMM) data for PMF analysis were obtained from Xie et al. (2022)
and are summarized in Table S3. Four- to ten-factor solutions were tested to retrieve a
final factor number with the most physically interpretable base-case solution.
**3. Results and discussion**
*3.1 Solvent test*
3.1.1 Extraction efficiency of different solvents
The concentrations of OC and EC fractions in each sample prior to solvent
extractions are listed in Table S1. SEOC concentrations and extraction efficiencies of
individual solvents and solvent mixtures are detailed in Table 1. Generally, DMF
presented the highest extraction efficiency of total OC whenever filter samples were
extracted once (89.0 ± 7.96%) or twice (95.6 ± 3.67%), followed by MeOH (one-time
extraction 82.3 ± 8.68%, two-time extraction 86.6 ± 7.86%) and MeOH/DCM mixtures
(~75%, ~85%). Although THF and DMF are frequently used to dissolve polymers (e.g.,
polystyrene) for characterization, THF had the lowest η values (64.2 ± 8.08%, 70.1 ±
8.01%) comparable to water (66.7 ± 8.58%, 69.9 ± 5.88%). Compared with one-time
extraction, the extraction efficiencies of selected solvents were improved by a few
percent when filter samples were extracted twice, and η values of MeOH/DCM
mixtures became closer to those of MeOH (Table 1). These results showed that solvents



can reach more than 80% of their dissolving capacity with the one-time extraction, and
the ambient OC in Nanjing is more soluble in MeOH than DCM.
From OC1 to OC4, the volatility of OC fractions is expected to decrease
continuously, and the molecules in OC fractions evolving at higher temperatures should
be larger than those in OC1 with similar functional groups. In Table 1, MeOH and
MeOH/DCM mixtures had comparable or even higher $\eta$ values (82.6 $\pm$ 25.9%–97.9 $\pm$
5.02%) of OC1 and OC2 than DMF (88.8 $\pm$ 4.98%–97.2 $\pm$ 2.12%). But OC3 and OC4
accounted for more than 60% of OC concentrations, and DMF exhibited significant ($p$
< 0.05) larger $\eta$ values than other solvents, indicating that DMF had stronger dissolving
capacity for large organic molecules than MeOH.
Concentrations of extracted OC fractions in MeOH, MeOH/DCM mixtures, and
THF based on the two methods for rOC measurements (*section 2.2*) are compared in
Figures S1 and S2. The total SEOC concentrations derived from the two methods are
compared in Figure S3. All the scatter data of SEOC fell along the 1:1 line with
significant correlations ($r > 0.85$, $p < 0.01$). Because the measurement uncertainty of
dominant species is lower than minor ones (Hyslop and White, 2008; Yang et al., 2021),
the slightly greater relative difference between the two methods for extractable OC1
was likely attributed to its low concentrations (< 1 $\mu$g m$^{-3}$; Tables 1 and S1). Thus,
baking extracted filters to dryness was expected to have little influence on SEOC
measurements, particularly for low-volatile OC fractions (OC2-OC4).
Although water dissolves less OC than MeOH, WSOC is intensively extracted and
analyzed for its composition and light absorption (Hecobian et al., 2010; Liu et al., 2013;
Washenfelder et al., 2015). WSOC can play a significant role in changing the radiative
and cloud-nucleating properties of atmospheric aerosols (Hallar et al., 2013; Taylor et
al., 2017). It also served as a proxy measurement for oxygenated (OOA) or secondary





organic aerosols (SOA) in some regions (Kondo et al., 2007; Weber et al., 2007). In
previous work, MeOH was commonly used as the most efficient solvent in extracting
OC from biomass burning ($\eta > 90\%$; Chen and Bond, 2010; Xie et al., 2017b) and
ambient particles ($\eta \sim 80\%$; Xie et al., 2019b; Xie et al., 2022). MeOH-insoluble OC
has rarely been investigated through direct solvent-extraction followed by instrumental
analysis. There is evidence showing that BrC absorption is associated mostly with large
molecular weight and extremely low-volatile species (Saleh et al., 2014; Di Lorenzo
and Young, 2016; Di Lorenzo et al., 2017). Compared with DMF, the lower capability
of MeOH in dissolving OC3 and OC4 would lead to an underestimation of BrC
absorption in atmospheric aerosols.
3.1.2 Light absorption of different solvent extracts
Table 2 shows the average $Abs_\lambda$ and $MAE_\lambda$ values of different solvent extracts at
365 and 550 nm. The $Abs_\lambda$ and $MAE_\lambda$ spectra of selected samples are illustrated in
Figure S4. Not including DMF, MeOH extracts exhibited the strongest light absorption.
Since MeOH can dissolve more OC3 and OC4 than DCM (Table 1), the $Abs_\lambda$ and $MAE_\lambda$
of MeOH/DCM extracts decreased as the fraction of DCM increased in solvent
mixtures (Table 2 and Figure S4). Water and THF extracts had the smallest $Abs_\lambda$ and
$MAE_\lambda$ due to their low extraction efficiencies for low-volatile OC (OC2-OC4; Table 1).
In comparison to MeOH extracts, $Abs_{365/550}$ and $MAE_{365/550}$ of DMF extracts were at
least more than 40% higher ($p < 0.05$). Given that the relative difference in extraction
efficiency of total OC between MeOH and DMF was less than 10%, low-volatile OC
should contain stronger light-absorbing chromophores (Saleh et al., 2014). Moreover,
the relative difference in $Abs_\lambda$ and $MAE_\lambda$ between MeOH and DMF extracts increased



with wavelength (Figure S4). This is because the light absorption of DMF extracts
depends less on wavelengths than other solvent extracts (Å ~4.5, Table 2).
In this work, insoluble organic particles coming off the filter during sonication
might lead to overestimated SEOC concentrations and η values, and then the $MAE_\lambda$ of
solvent extracts would be underestimated. Previous studies rarely considered the loss
of insoluble OC during the extraction process (Yan et al., 2020), of which the impact
on $MAE_\lambda$ calculation was still inconclusive. But $Abs_\lambda$ measurements would never be
influenced, as the light absorbance of solvent extracts was analyzed after filtration. In
Table 2, the second extraction only increases the average $Abs_{365}$ and $Abs_{550}$ values of
DMF extracts by 6.70% ($p = 0.78$) and 6.76% ($p = 0.77$), respectively. We suspected
that the difference in η values of DMF between one-time and two-time extraction
procedures was mainly ascribed to the detachment of insoluble OC particles.
In Figure S5, the UV/Vis spectra of 4-nitrophenol and 4-nitrocatechol in DMF are
very different from other solvents with maximum absorbance at ~450 nm, indicating
that the solvent type should influence solution absorption. However, the absorbance of
4-nitrophenol and 4-nitrocatechol in DMF at 365 nm ($A_{365}$) was lower than that in
MeOH, and PAH solutions showed very similar absorbance spectra across the five
solvents (Figure S5g–l and Table S4). Considering that low-volatile OC fractions (e.g.,
OC3 and OC4) in the ambient are less water soluble (Table 1) and have a high degree
of conjugation (Chen and Bond, 2010; Lin et al., 2014), their structures are probably
featured by a PAH skeleton. Therefore, the large difference in $Abs_{365}$ between DMF and
MeOH extracts (Table 2) was primarily ascribed to the fact that DMF can dissolve more
OC3 and OC4 than methanol (Table 1), but not the solvent effect.





*3.2 Collocated measurements and temporal variability*
$Abs_{365}$ values of collocated $Q_f$ and $Q_b$ extracts in DMF are summarized in Table S5.
No significant difference was observed ($Q_f$ $p = 0.96$; $Q_b$ $p = 0.42$) between the two
samplers. After $Q_b$ corrections, $Abs_{365}$, $MAE_{365}$, and Å of DMF extractable OC ($Abs_{365,d}$,
$MAE_{365,d}$, and $Å_d$) in $PM_{2.5}$ were calculated by averaging each pair of duplicate $Q_f$–$Q_b$
data, and are compared with those of methanol extracts ($Abs_{365,m}$, $MAE_{365,m}$, and $Å_m$)
in Table 3. Figure 1 shows comparisons between collocated measurements of $Abs_{365,d}$,
$MAE_{365,d}$, and $Å_d$. Generally, all comparisons indicated good agreement with COD <
0.20 (0.094–0.15). $Abs_{365,d}$ and $MAE_{365,d}$ had comparable uncertainty fractions (ARPD,
22.7% and 24.5%, Figure 1) as $Abs_{365,m}$ and $MAE_{365,m}$ (28.4% and 28.8%; Xie et al.,
2022). Since different primary combustion sources can have similar spectral
dependence for BrC absorption (Chen and Bond, 2010; Xie et al., 2017b; Xie et al.,
2018; Xie et al., 2019a), most $Å_d$ data clustered on the identity line with much lower
variability than $Abs_{365,d}$ and $MAE_{365,d}$. As shown in Table 3, average $Abs_{365,d}$ and
$MAE_{365,d}$ values were 30.7% ($p < 0.01$) and 17.3% ($p < 0.05$) larger than average
$Abs_{365,m}$ and $MAE_{365,m}$. Because the $k$ value of BrC in bulk solution is directly estimated
from $Abs_λ$ or $MAE_λ$ (Liu et al., 2013; Liu et al., 2016; Lu et al., 2015), the estimation
method needs to be revised when ambient BrC is extracted using DMF instead of
MeOH. In comparison to $Å_m$ (6.81± 1.64; Table 3), the lower average $Å_d$ (5.25 ± 0.64,
$p < 0.01$) supports that more-absorbing BrC had less spectral dependence than less-
absorbing BrC.
Figure 2 compares the time series of $Abs_{365}$, $MAE_{365}$, and Å between the DMF and
MeOH extracts. Both DMF and MeOH extracts had significant ($p < 0.05$) higher
absorption at night-time than during the daytime due to the "photo-bleaching" effect
(Zhang et al., 2020; Xie et al., 2022). All the three parameters of DMF and MeOH





extracts exhibited consistency in winter (Figure 2) when biomass burning dominated
BrC absorption (Xie et al., 2022). While in later spring and summer (2019/05/15–
2019/08/01), average $Abs_{365,d}$ and $MAE_{365,d}$ values were more than two times greater
than the average $Abs_{365,m}$ and $MAE_{365,m}$. Many studies have identified a temporal
pattern of BrC absorption with winter maxima and summer minima based on
water/MeOH extraction methods (Lukács et al., 2007; Zhang et al., 2010; Du et al.,
2014; Zhu et al., 2018). Due to the low capability of water and MeOH in dissolving
large BrC molecules, BrC absorption and its temporal variations in these studies might
be biased. Moreover, the identification of BrC sources using receptor models is highly
dependent on the difference in the time series of input species (Dall'Osto et al., 2013).
Then, using DMF instead of MeOH for BrC extraction and measurements will lead to
distinct source apportionment results.
*3.3 Sources of DMF and MeOH Extractable BrC*
A final factor number of eight was determined based on the interpretability of
different base-case solutions (four to ten factors). Normalized factor profiles of seven-
to nine-factor solutions are compared in Figure S6. The seven-factor solution failed to
resolve the lubricating oil combustion factor characterized by hopanes and steranes
(Figure S6c). An unknown factor containing various source tracers related to crustal
dust ($Ca^{2+}$ and $Mg^{2+}$), lubricating oil (hopanes and steranes), and soil microbiota (sugar
and sugar alcohols) was identified in the nine-factor solution (Figure S6i). Median and
mean values of input $Abs_{365,d}$, $Abs_{365,m}$ and bulk component concentrations agreed well
with PMF estimations (Table S6), and the strong correlations ($r = 0.86$–$0.99$) between
observations and PMF estimations indicated that the eight-factor solution simulated the
time series of input species well. In comparison to Xie et al. (2022), where $Abs_{365}$ of
MeOH and water extracts were apportioned to nine sources using the same speciation



data, this work lumped secondary nitrate and sulfate to the same factor (termed
"secondary inorganics", Figure S6h), and the other seven factors had similar factor
profiles linked with biomass burning, non-combustion fossil, lubricating oil
combustion, coal combustion, dust resuspension, biogenic emission, and isoprene
oxidation. Interpretations of individual factors based on characteristic source tracers
and contribution time series were provided in previous work (Gou et al., 2021; Xie et
al., 2022). The average relative contributions of the identified factors to $Abs_{365,d}$,
$Abs_{365,m}$, and bulk components are listed in Table S7. Consistent contribution
distributions of $Abs_{365,m}$ were observed between Xie et al. (2022) and this study,
indicating that the PMF results were robust to the inclusion of $Abs_{365,d}$ data. Figure 3
compares the time series of factor contributions to $Abs_{365,d}$ and $Abs_{365,m}$. Although
$Abs_{365,d}$ and $Abs_{365,m}$ had comparable contributions from biomass burning, lubricating
oil combustion, and coal combustion (Figure 3a, c, d), other sources had significant ($p$
$< 0.01$) higher average contributions to $Abs_{365,d}$ than $Abs_{365,m}$.
The non-combustion fossil factor represents unburned fossil-fuel emissions (e.g.,
petroleum products), which contain substantial large organic molecules (e.g., high MW
PAHs; Simoneit and Fetzer, 1996; Mi et al., 2000). This might explain why the non-
combustion fossil factor contributed more $Abs_{365,d}$ than $Abs_{365,m}$ all over the year. Dust
resuspension and isoprene oxidation factors show prominent contributions to $Abs_{365,d}$
in spring and summer, respectively (Figure 3e, g). The dust resuspension factor had the
highest average contributions to both crustal materials ($Ca^{2+}$ and $Mg^{2+}$) and
carbonaceous species (OC and EC; Table S7 and Figure S6), and was considered a
mixed source of crustal dust and motor vehicle emissions (Yu et al., 2020; Xie et al.,
2022). Besides the influences from primary emissions, aging processes of organic
components in dust aerosols can induce the formation of BrC through iron-catalyzed



polymerization (Link et al., 2020; Al-Abadleh, 2021; Chin et al., 2021). It was
demonstrated that the isoprene-derived polymerization products through aerosol-phase
reactions are light-absorbing chromophores (Lin et al., 2014; Nakayama et al., 2015).
The biogenic emission factor was characterized by tracers related to microbiota
activities (sugar and sugar alcohols) and decomposition of high plant materials (odd-
numbered alkanes) in soil (Rogge et al., 1993; Simoneit et al., 2004), and had negligible
contributions ($< 0.1\%$) to $Abs_{365,d}$ and $Abs_{365,m}$. Evidence shows that secondary BrC
can be generated through gas-phase reactions of anthropogenic volatile organic
compounds with $NO_X$ (Nakayama et al., 2010; Liu et al., 2016; Xie et al., 2017a),
aqueous reactions of SOA with reduced nitrogen-containing species (e.g., $NH_4^+$;
Updyke et al., 2012; Powelson et al., 2014; Lin et al., 2015), and evaporation of water
from droplets in the atmosphere containing soluble organics (Nguyen et al., 2012;
Kasthuriarachchi et al., 2020). Their contributions might be lumped into the secondary
inorganics factor due to the lack of OMMs. According to these results, one possible
explanation for the difference in time series between $Abs_{365,d}$ and $Abs_{365,m}$ (Figure 2) is
that large BrC molecules from unburned fossil fuels and atmospheric processes are less
soluble in MeOH than DMF.
**4 Conclusions and implications**

The comparisons of extraction efficiencies of ambient OC across selected solvents

and solvent mixtures reveal the necessity of replacing MeOH with DMF for measuring
BrC absorption in ambient aerosols, as low-volatile OC fractions containing strong
chromophores are less soluble in MeOH than DMF. The light-absorption measurements
of different solvent extracts show that DMF can extract more light-absorbing materials
from ambient aerosols than MeOH. Existing modeling studies on the radiative forcing
of BrC (Feng et al., 2013; Wang et al., 2014; Zhang et al., 2020) often retrieved or



estimated its optical properties from laboratory or ambient measurements based on
water/methanol extraction methods (Chen and Bond, 2010; Hecobian et al., 2010; Liu
et al., 2013; Zhang et al., 2013), and probably underestimated the contribution of BrC
to total aerosol absorption.
Although light-absorbing properties of DMF and MeOH extracts had good
agreement in cold periods, their distinct time series in spring and summer implies that
the contributions of certain BrC sources were underestimated or missed when the
MeOH extraction method was used. Source apportionment results of $Abs_{365,d}$ and
$Abs_{365,m}$ based on organic molecular marker data indicated that large and methanol
insoluble BrC molecules are likely coming from unburned fossil fuels and
polymerization of aerosol organics. Laboratory studies have observed the
polymerization process through heterogeneous reactions of several precursors (e.g.,
catechol; Lin et al., 2014; Link et al., 2020), but the structures and light-absorbing
properties of polymerization products in ambient aerosols are less understood and
warrant further study.

***Data availability***
Data used in the writing of this paper is available at the Harvard Dataverse
(https://doi.org/10.7910/DVN/CGHPXB, Xu et al., 2022)

***Author contributions***
MX designed the research. ZX, WF, YW, and HY performed laboratory experiments.
ZX, WF, and MX analyzed the data. ZX and MX wrote the paper with significant
contributions from YW and HL.





*Competing interests*

The authors declare that they have no conflict of interest.

*Acknowledgments*

This work was supported by the National Natural Science Foundation of China

(NSFC, 42177211, 41701551).

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



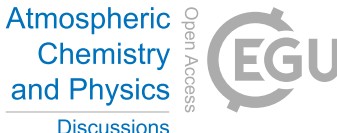

Table 1. SEOC concentrations and extraction efficiencies (η, %) of total OC and OC fractions for different solvents.

| | OC prior to extractions | Water[a] | MeOH[b] | MeOH/DCM (1:1)[b] | MeOH/DCM (1:2)[b] | THF[b] | DMF[a] |
|---|---|---|---|---|---|---|---|
| **One-time extraction ($N$ = 11)** | | | | | | | |
| *SEOC, µg m⁻³* | | | | | | | |
| Total OC | 9.36 ± 2.27 | 6.38 ± 2.03 | 7.85 ± 2.40 | 7.08 ± 1.32 | 6.99 ± 1.71 | 6.14 ± 2.01 | 8.49 ± 2.52 |
| OC1 | 0.66 ± 0.21 | 0.61 ± 0.20 | 0.64 ± 0.21 | 0.65 ± 0.20 | 0.64 ± 0.22 | 0.59 ± 0.18 | 0.59 ± 0.24 |
| OC2 | 2.69 ± 0.55 | 2.20 ± 0.60 | 2.50 ± 0.55 | 2.34 ± 0.41 | 2.37 ± 0.46 | 2.09 ± 0.55 | 2.48 ± 0.60 |
| OC3 | 3.35 ± 0.93 | 1.82 ± 0.80 | 2.48 ± 0.96 | 2.23 ± 0.49 | 2.18 ± 0.70 | 1.98 ± 0.93 | 2.86 ± 1.01 |
| OC4 | 2.75 ± 0.81 | 1.76 ± 0.65 | 2.23 ± 0.84 | 1.86 ± 0.51 | 1.78 ± 0.61 | 1.48 ± 0.61 | 2.56 ± 0.87 |
| *η (%)* | | | | | | | |
| Total OC | | 66.7 ± 8.58 | 82.3 ± 8.68 | 76.0 ± 7.70 | 74.3 ± 7.83 | 64.2 ± 8.08 | 89.0 ± 7.96 |
| OC1 | | 91.7 ± 4.85 | 96.1 ± 6.73 | 97.9 ± 5.02 | 97.4 ± 4.35 | 89.6 ± 9.55 | 88.8 ± 4.98 |
| OC2 | | 80.8 ± 8.11 | 92.7 ± 3.69 | 87.7 ± 5.87 | 88.5 ± 7.21 | 76.9 ± 7.62 | 91.4 ± 6.17 |
| OC3 | | 52.4 ± 11.8 | 73.0 ± 11.5 | 68.1 ± 8.64 | 65.2 ± 10.2 | 57.6 ± 12.0 | 84.3 ± 9.79 |
| OC4 | | 63.3 ± 9.13 | 80.3 ± 11.4 | 69.0 ± 9.26 | 64.5 ± 8.11 | 52.7 ± 5.86 | 92.8 ± 9.69 |
| **Two-time extraction ($N$ = 10)** | | | | | | | |
| *SEOC, µg m⁻³* | | | | | | | |
| Total OC | 10.9 ± 4.93 | 7.74 ± 4.01 | 9.33 ± 4.11 | 9.34 ± 4.19 | 9.11 ± 4.04 | 7.56 ± 3.38 | 10.4 ± 4.80 |
| OC1 | 0.66 ± 0.47 | 0.62 ± 0.45 | 0.62 ± 0.49 | 0.59 ± 0.50 | 0.60 ± 0.51 | 0.59 ± 0.49 | 0.60 ± 0.47 |
| OC2 | 2.76 ± 0.77 | 2.20 ± 0.59 | 2.60 ± 0.66 | 2.57 ± 0.65 | 2.60 ± 0.68 | 2.28 ± 0.53 | 2.69 ± 0.78 |
| OC3 | 4.11 ± 2.01 | 2.55 ± 1.62 | 3.26 ± 1.62 | 3.37 ± 1.68 | 3.20 ± 1.58 | 2.62 ± 1.39 | 3.88 ± 1.95 |
| OC4 | 3.36 ± 1.77 | 2.38 ± 1.42 | 2.84 ± 1.42 | 2.81 ± 1.47 | 2.71 ± 1.39 | 2.08 ± 1.06 | 3.23 ± 1.70 |
| *η (%)* | | | | | | | |
| Total OC | | 69.9 ± 5.88 | 86.6 ± 7.86 | 86.2 ± 8.73 | 84.8 ± 7.76 | 70.1 ± 8.01 | 95.6 ± 3.67 |
| OC1 | | 93.6 ± 4.08 | 90.3 ± 13.9 | 82.6 ± 25.9 | 83.8 ± 22.4 | 82.9 ± 15.1 | 92.2 ± 13.9 |
| OC2 | | 80.1 ± 5.01 | 94.8 ± 4.20 | 93.6 ± 4.94 | 94.7 ± 2.51 | 83.5 ± 6.86 | 97.2 ± 2.12 |
| OC3 | | 59.0 ± 10.6 | 80.0 ± 10.2 | 82.3 ± 9.86 | 79.1 ± 10.6 | 63.9 ± 10.7 | 94.2 ± 4.15 |
| OC4 | | 69.3 ± 6.46 | 86.3 ± 12.0 | 84.3 ± 12.0 | 82.7 ± 13.3 | 62.9 ± 7.76 | 96.9 ± 5.18 |

[a] Concentrations of rOC in extracted filters were measured after the baking process (100 ℃, 2 h); [b] rOC was measured when extracted filters were air dried.





Table 2. Light-absorbing properties of SEOC following one-time and two-time extraction procedures.

| Solvent | Water | MeOH | MeOH/DCM (1:1) | MeOH/DCM (1:2) | THF | DMF |
|---|---|---|---|---|---|---|
| **One-time extraction** | | | | | | |
| $Abs_{365}$, $Mm^{-1}$ | 5.13 ± 2.04 | 11.9 ± 5.83 | 10.3 ± 4.42 | 8.12 ± 3.38 | 5.48 ± 3.01 | 17.5 ± 8.05 |
| $Abs_{550}$, $Mm^{-1}$ | 0.35 ± 0.12 | 1.28 ± 0.87 | 0.97 ± 0.55 | 0.35 ± 0.47 | 0.42 ± 0.47 | 4.40 ± 2.34 |
| $MAE_{365}$, $m^2\ g^{-1}$ C | 0.87 ± 0.19 | 1.46 ± 0.41 | 1.41 ± 0.36 | 1.13 ± 0.22 | 0.87 ± 0.25 | 2.02 ± 0.58 |
| $MAE_{550}$, $m^2\ g^{-1}$ C | 0.062 ± 0.028 | 0.15 ± 0.084 | 0.13 ± 0.054 | 0.042 ± 0.52 | 0.059 ± 0.56 | 0.30 ± 0.12 |
| Å | 6.63 ± 0.49 | 5.44 ± 0.75 | 5.65 ± 0.54 | 6.59 ± 0.66 | 6.17 ± 0.69 | 4.52 ± 0.41 |
| **Two-time extraction** | | | | | | |
| $Abs_{365,1st}$,[a] $Mm^{-1}$ | 6.64 ± 4.25 | 14.1 ± 7.09 | 14.6 ± 8.05 | 11.6 ± 6.78 | 7.17 ± 4.26 | 20.5 ± 10.6 |
| $Abs_{550,1st}$,[a] $Mm^{-1}$ | 0.42 ± 0.12 | 1.34 ± 0.70 | 1.34 ± 0.83 | 0.84 ± 0.50 | 0.53 ± 0.27 | 2.82 ± 1.44 |
| $Abs_{365}$,[b] $Mm^{-1}$ | 8.26 ± 5.21 | 15.5 ± 7.76 | 16.8 ± 8.82 | 14.0 ± 8.91 | 8.35 ± 4.81 | 21.9 ± 11.2 |
| $Abs_{550}$,[b] $Mm^{-1}$ | 0.50 ± 0.18 | 1.60 ± 0.78 | 1.64 ± 0.99 | 1.22 ± 0.98 | 0.69 ± 0.43 | 3.01 ± 1.49 |
| $MAE_{365}$, $m^2\ g^{-1}$ C | 1.19 ± 0.26 | 1.70 ± 0.60 | 1.80 ± 0.52 | 1.50 ± 0.51 | 1.10 ± 0.40 | 2.11 ± 0.49 |
| $MAE_{550}$, $m^2\ g^{-1}$ C | 0.082 ± 0.30 | 0.19 ± 0.11 | 0.17 ± 0.083 | 0.13 ± 0.069 | 0.094 ± 0.054 | 0.29 ± 0.075 |
| Å | 6.32 ± 0.58 | 5.37 ± 0.57 | 5.47 ± 0.67 | 5.57 ± 0.39 | 6.06 ± 0.54 | 4.53 ± 0.21 |

[a] Light absorption coefficient of SEOC after the first extraction; [b] sum of SEOC absorption in 1st and 2nd extracts.





Table 3. Comparisons of light-absorbing properties of ambient $PM_{2.5}$ extracts in DMF and MeOH derived from duplicate $Q_f$–$Q_b$ data ($N = 109$).

| | DMF | | | MeOH[a] | | |
|---|---|---|---|---|---|---|
| | Median | Mean ± std | Range | Median | Mean ± std | Range |
| $Abs_{365}$, Mm-1 | 6.99 | 8.42 ± 5.40 | 1.14–30.8 | 5.59 | 6.43 ± 4.66 | 0.38–29.6 |
| $MAE_{365}$, m$^2$ g$^{-1}$C | 1.13 | 1.20 ± 0.49 | 0.34–2.45 | 0.91 | 1.03 ± 0.58 | 0.089–2.49 |
| Å | 5.21 | 5.25 ± 0.64 | 3.21–6.82 | 6.49 | 6.81 ± 1.64 | 4.34–11.3 |

[a] Data for MeOH extracts were obtained from Xie et al. (2022).



Figure 1

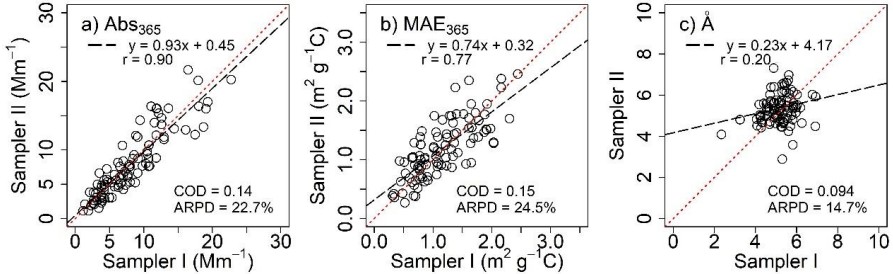

Figure 1. Comparisons between collocated measurements for light-absorbing properties of PM$_{2.5}$ extracts in DMF after Q$_b$ corrections.





Figure 2

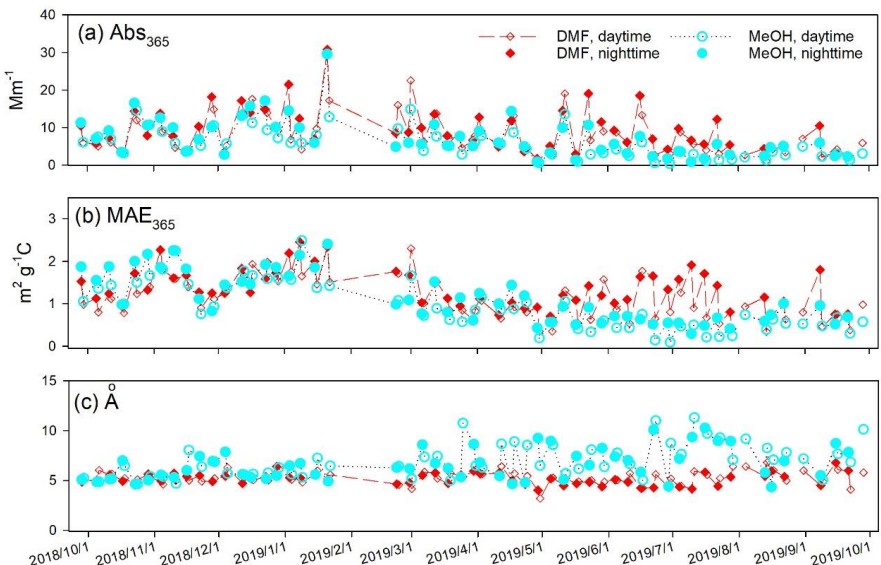

Figure 2. Time series comparisons of light-absorbing properties of DMF and MeOH extracts using artifact-corrected data. MeOH extract data were obtained from Xie et al. (2022).



## Figure 3

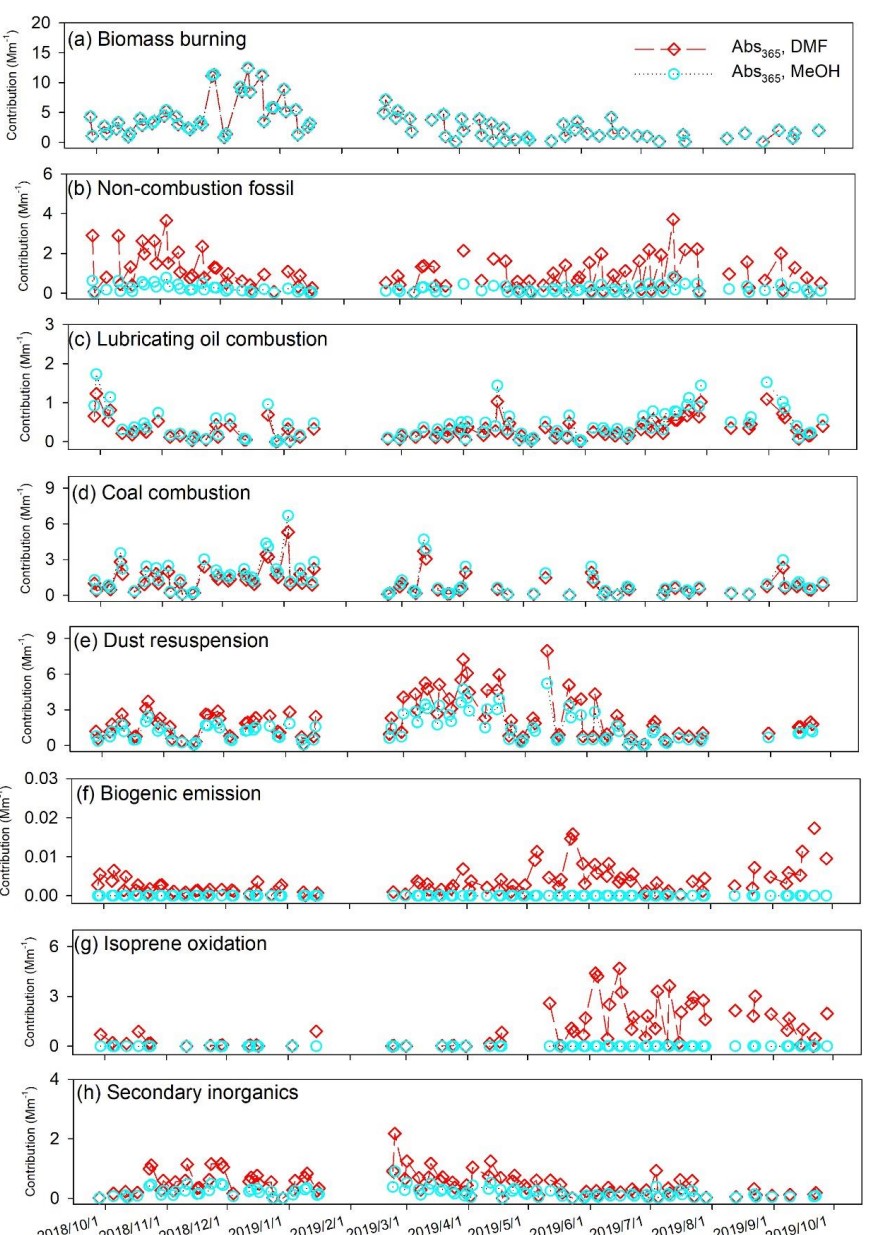

Figure 3. Time series of factor contributions to Abs$_{365}$ of DMF and MeOH extracts of ambient PM$_{2.5}$ samples.