# Peer review of "Potential underestimation of ambient brown carbon absorption based on the methanol extraction method and its impacts on source analysis Zhenqi Xua, Wei Fenga, Yicheng Wanga, Haoran Yea, Yuhang Wangb, Hong Liaoa, Mingjie Xiea,\*</"

_Atmospheric Chemistry and Physics, 2022_

## Referee Comment (RC3)

In this manuscript, the authors investigate the light absorption characteristics of solvent-extractable brown carbon aerosol in ambient samples affected by multiple sources. The manuscript raises the topic of the effect of the selected solvent, or solvent mixture, on the extraction efficiency of organics and subsequently on the absorbance measured by UV-visible spectroscopy and associated light absorption properties (mass-normalized absorbance and its wavelength dependence). They find that N, N-dimethylformamide dissolves BrC associated with unburned fossil fuel and polymerization processes of aerosol organics more efficiently than methanol. The study results and potential implications are of interest to readers of *Atmospheric Chemistry and Physics.* Nevertheless, I would only recommend publication of this manuscript upon careful consideration by the authors of the specific comments below and subsequent clarifications and fine-tuning of some discussions in the revised manuscript.

**Specific comments**

Line 1: the message of the title may not be universal and could be considered misleading. If the authors prefer the current title to largely remain (although Reviewer 1 has already provided an alternative that may better reflect the paper content), I strongly recommend they change it to: "Potential underestimation…". The main reason is that methanol still efficiently extracted biomass burning-related BrC, which has been more widely studied in the literature.

Lines 44-45: for completeness, please clarify the following details in the abstract: (1) BrC aerosols associated with biomass burning, and coal (?), combustion sources were still highly soluble in MeOH; (2) the different MeOH solubility of BrC from different (seasonal) sources was likely the main reason for the aforementioned distinct time series. Therefore, (3) a more accurate alternative to the sentence: "These results highlight the necessity of replacing MeOH with DMF for further investigations on structures…" could be: "These results highlight the importance of testing different solvents to investigate the structures and light absorption of BrC, particularly of the low-volatility fraction potentially associated with certain non-traditional sources.". Please also rephrase related statements in Lines 103-104, 105-107, 227-228 ("potential underestimation"), 415, 418 ("may sometimes"), 423-424 ("…may potentially underestimate the contribution of solvent-extractable…"), and elsewhere if applicable.

Line 93: the sentence is more informative with the following (or similar) addition referring to the extractable aerosol fraction: "…directly if the latter is not converted to particulate absorption with Mie calculations, solvent-matrix, and pH effects are not accounted for, and solvent solubility is not high.".

Line 120: that is likely true for DMF; THF has been tested for biomass burning-influenced ambient BrC (Moschos et al., 2021); do the two observations agree? The authors state "rarely" in this sentence: do any other studies exist that have tested any of these two solvents for extracting BrC aerosol?

Line 195: when measuring the absorbance of solvent extracts, solvent-matrix effects (Reichardt, 2003) are not uncommon (yet rarely accounted for in the BrC research). Chen and Bond (2010; cited in the preprint), Mo et al. (2017), and Moschos et al. (2021) observed higher absorbance of water-extracted BrC aerosol that was further diluted/re-dissolved in methanol (for the same total extract volume). Could the authors discuss, in the revised manuscript, similar effects for their selected solvents/mixtures, as well as the implications for the results presented here when not correcting for such (i.e., currently, the solvent-matrix vs. solubility effects are not decoupled)? Can the authors rule out a solvent-matrix effect that would affect the wavelength-dependent comparison between MeOH and DMF, for example, in Fig. S4?

Line 312: here, the authors have the opportunity to discuss also potential pH effects, e.g., the absorbance red-shift for the water-extract of 4-nitro-catechol in Fig. S5 and the isosbestic point ~365 nm, which seem to be consistent with the observation of Lin et al. (2017) for water vs. organic-solvent BrC aerosol extracts.

Line 426: please clarify: "…in cold periods, when coal/biomass burning sources dominated the aerosol emissions..." if that is the case.

Conclusions and implications section: It is important here to provide a broader view that will allow future studies to confirm these observations, while other approaches may still be helpful: for example, the authors could state that a combination of solvents with a broad polarity index (e.g., Lin et al., 2018) may still be good choice to cover different conditions, e.g., a mixture of non-polar (e.g., hexane), polar protic (e.g., MeOH) and polar aprotic solvents (e.g., DMF) for a range of BrC-containing samples influenced by different sources. Based on this and other comments above, there is no "universal evidence" from this study that DMF is the unique-best solvent for BrC under all conditions. At the same time, a more balanced discussion in the revised manuscript would encourage future studies to test DMF and potentially verify or revise the authors' observations.

Line 434: does that refer to the isoprene oxidation factor in Fig. 3? That is an important finding; the figure can be cited once more in this section together with this statement.

Figure 3: please mention (possibly in the caption) that the biogenic emission factor *Abs* is below the detection limit (if that is the case). Further, I agree with Reviewer 2 that it is critical to provide evidence for the robustness of the PMF solution for a reader to assess the quality of the results and the validity of the associated conclusions. Could the authors also discuss the yearly evolution of the *Abs* relative difference between the two solvents for each PMF factor in Fig. 3? The relative difference seems low for coal and biomass burning throughout the year; what are the time-series trend and day-to-day variability for the other factors (those where both solvents seem to dissolve a non-negligible fraction of their chemical constituents)? Finally, based on the statement in Lines 297-299, could the authors reproduce Fig. 3 for *Abs* at a longer wavelength and compare the two PMF results?

**Technical corrections**

Lines 281, 295, 315 & 416: "low-volatility".
Table 3: correct the superscript to "$Mm^{-1}$".
Table S3: please provide the units of the tabulated data other than $Abs_{365}$.

**References**

Lin, P., Bluvshtein, N., Rudich, Y., Nizkorodov, S. A., Laskin, J., and Laskin, A.: Molecular chemistry of atmospheric brown carbon inferred from a nationwide biomass burning event, Environ. Sci. Technol., 51, 11561–11570, https://doi.org/10.1021/acs.est.7b02276, 2017.

Lin, P., Fleming, L. T., Nizkorodov, S. A., Laskin, J., and Laskin, A.: Comprehensive molecular characterization of atmospheric brown carbon by high resolution mass spectrometry with electrospray and atmospheric pressure photoionization, Anal. Chem., 90, 12493–12502, https://doi.org/10.1021/acs.analchem.8b02177, 2018.

Mo, Y., Li, J., Liu, J., Zhong, G., Cheng, Z., Tian, C., Chen, Y., and Zhang, G.: The influence of solvent and pH on determination of the light absorption properties of water-soluble brown carbon, Atmos. Environ., 161, 90–98, https://doi.org/10.1016/j.atmosenv.2017.04.037, 2017.

Moschos, V., Gysel-Beer, M., Modini, R. L., Corbin, J. C., Massabò, D., Costa, C., Danelli, S. G., Vlachou, A., Daellenbach, K. R., Szidat, S., Prati, P., Prévôt, A. S. H., Baltensperger, U., and El Haddad, I.: Source-specific light absorption by carbonaceous components in the complex aerosol matrix from yearly filter-based measurements, Atmos. Chem. Phys., 21, 12809–12833, https://doi.org/10.5194/acp-21-12809-2021, 2021.

Reichardt, C.: Solvents and solvent effects in organic chemistry, 3 Edn., Wiley-VCH Verlag GmbH & Co. KGaA, 329–388, 2003.

---

## Author Comment (AC1)

Xu et al. examined the influence of solvent selection on brown carbon (BrC) absorption measurements and source analysis for ambient aerosols. Water, methanol, methanol-DCM mixtures, THF, and DMF were tested. Measurement results showed that DMF exhibited the highest extraction efficiency of ambient organic carbon (OC), particularly for low-volatile OC, and DMF extracts also had significant higher light absorption than other solvent extracts. Moreover, the comparison of sources between DMF and methanol extract absorption is very interesting and indicates that the methanol-extraction method will underestimate BrC contributions from non-combustion sources. The authors suggested that DMF can extract more BrC than commonly used solvents. DMF might be an important solvent for investigating low-volatile OC in the near future. This manuscript provides very useful information for further studies on radiative forcing and sources of organic aerosols, and I recommend the publication of this manuscript in ACP, though I'd like the authors to address some minor specific comments below.

**1.** In this work, several solvent extracts of ambient OC were measured for light absorption, would the authors consider changing the title to "The dependence of brown carbon absorption on solvent selection and its impacts on source analysis", or something similar to highlight the differences in different solvent extraction methods?

*Reply:*

In the current work, we examined the difference in the extraction efficiency and light absorption of solvent-extractable OC across five solvents and solvent mixtures. Only the solvent with the highest extraction efficiency (N, N-dimethylformamide, DMF) was applied to extract a matrix of ambient $PM_{2.5}$ samples for light absorption measurement. In comparison to methanol (MeOH)-extractable OC, DMF extracts showed significant ($p < 0.01$) higher light absorption, and the light absorption of methanol-insoluble OC is mainly linked with unburned fossil fuel and polymerization processes of aerosol organics. Other solvents or solvent mixtures (water, MeOH/DCM, and THF) were not examined for source apportionment of BrC absorption.

Reviewer 3 suggests adding "potential" to the original title of the manuscript, as methanol can extract biomass burning BrC efficiently (> 90%). The ambient OC is a complex mixture coming from both primary and secondary sources. In this work, only ambient OC was extracted using different solvents. The methanol extraction method had a lower extraction efficiency than the DMF extraction method, and underestimated ambient BrC absorption. In the revised manuscript, we changed the title to

"Potential underestimation of ambient brown carbon absorption based on the methanol extraction method and its impacts on source analysis"

**2.** Line 31. "However, undissolved OC fractions will lead to underestimated BrC absorption." What is the magnitude of this underestimation? Also, what about the mass? If the undissolved fraction has low light absorption, the underestimation might not be large, right?

*Reply:*

Here, we only mentioned a potential problem associated with the methanol extraction method. As we stated in the introduction (lines 101–103, 108–113)

"The light absorption of biomass burning OM is majorly contributed by large molecules (MW > 500~1000 Da; Di Lorenzo and Young, 2016; Di Lorenzo et al., 2017) and depends on burn conditions (Saleh et al., 2014)."

"Methanol can extract > 90% OM from biomass burning (Chen and Bond, 2010; Xie et al., 2017b), while the extraction efficiency ($\eta$, %) decreases to ~80% for ambient organic aerosols (Xie et al., 2019b; Xie et al., 2022) possibly due to other sources emitting large hydrophobic molecules and oligomerizations of small molecules during the aging process (Cheng et al., 2021; Li et al., 2021). The light-absorbing properties and structures of methanol-insoluble OC (MIOC) are still unknown."

In this work, we demonstrated that DMF can extract more low-volatility OC from ambient OC than MeOH, and the MeOH-insoluble OC contained strong light-absorbing chromophores. These results have already been included in the abstract.

"Among the five solvents and solvent mixtures, DMF dissolved the highest fractions of ambient OC (up to ~95%), followed by MeOH and MeOH/DCM mixtures (< 90%), and the DMF extracts had significant ($p < 0.05$) higher light absorption than other solvent extracts. This is because the OC fractions evaporating at higher temperatures (> 280℃) are less soluble in MeOH (~80%) than in DMF (~90%) and contain stronger light-absorbing chromophores." (Lines 35–40)

**3.** Lines 41-42, "the light absorption of DMF and MeOH extracts of collocated aerosol samples in Nanjing showed distinct time series. Specifically, what is the difference, and do they have any common temporal patterns?

*Reply:*

In the revised manuscript, we specified the difference by changing the original expression to

"Moreover, the light absorption of DMF and MeOH extracts of collocated aerosol samples in Nanjing showed consistent temporal variations in winter when biomass burning dominated BrC absorption. While the average light absorption of DMF extracts was more than two times greater than the MeOH extracts in late spring and summer." (Lines 40–44)

**4.** Lines 58-60, "The radiative forcing (RF) of the light-absorbing organic carbon, also termed "brown carbon" (BrC), is not well quantified due to the lack of its emission data and large uncertainties in *in situ* BrC measurements" The secondary formation will also add complexity on RF estimation of BrC. Please mention it.

*Reply:*

Thanks, the original expression has been changed to

"The radiative forcing (RF) of the light-absorbing organic carbon, also termed "brown carbon" (BrC), is not well quantified due to the lack of its emission data,

complex secondary formations, and large uncertainties in *in situ* BrC measurements (Wang et al., 2014; Wang et al., 2018; Saleh, 2020)." (Lines 83–86)

**5.** Lines 261-262, "THF based on the two methods for rOC measurements (*section 2.2*) are compared in Figures S1 and S2." Would the authors consider putting these two figures in the main text? They provide very useful information.

*Reply:*

In this study, filters extracted using MeOH, MeOH/DCM (1:1), MeOH/DCM (1:2), and THF were air-dried in a fume hood and analyzed for residual OC (rOC, $\mu g$ $m^{-3}$). Filters extracted in water and DMF cannot be air-dried in the short term due to the low volatility of solvents, and their rOC was measured after baking at 100 °C for 2 h. To examine if the baking process would influence rOC measurements, the rOC of filters extracted in MeOH, MeOH/DCM mixtures, and THF were also measured after the baking process and compared with those determined after air drying (*Section 2.3*).

The results shown in Figures S1 and S2 indicate that baking extracted filters to dryness would have little influence on SEOC measurements (Lines 299–308).

"Concentrations of extracted OC fractions in MeOH, MeOH/DCM mixtures, and THF based on the two methods for rOC measurements (*section 2.2*) are compared in Figures S1 and S2. The total SEOC concentrations derived from the two methods are compared in Figure S3. All the scatter data of SEOC fell along the 1:1 line with significant correlations ($r > 0.85$, $p < 0.01$). Because the measurement uncertainty of dominant species is lower than minor ones (Hyslop and White, 2008; Yang et al., 2021), the slightly greater relative difference between the two methods for extractable OC1 was likely attributed to its low concentrations ($< 1$ $\mu g$ $m^{-3}$; Tables 1 and S1). Thus, baking extracted filters to dryness was expected to have little influence on SEOC measurements, particularly for low-volatility OC fractions (OC2-OC4)."

These two figures were only used to validate rOC measurements for filters extracted in water and DMF, and were not cited elsewhere or directly linked with the main topic of the manuscript. Therefore, we kept these two figures in the supplementary information.

**6.** Section 3.1.2. Is the difference across solvent extraction methods related to the physicochemical properties of OC? If it is true, please state which factors have a substantial influence.

*Reply:*

According to the results provided in Tables 1 and 2, DMF and MeOH (or MeOH/DCM mixtures) had comparable extraction efficiencies for more volatile OC (OC1 and OC2). However, DMF exhibited significant ($p < 0.05$) higher efficiency in extracting low-volatility OC (OC3 and OC4) than other solvents, and the low-volatility OC accounted for more than 60% of OC concentrations. This is expected to be the main reason for the fact that the light absorption of DMF extracts was significantly ($p < 0.05$)

higher than other solvent extracts, as low-volatility OC contains stronger light-absorbing chromophores (Saleh et al., 2014).

The difference in the light absorption across solvent extraction methods might depend on the fraction of low-volatility OC. In another word, the difference will increase as the fraction of low-volatility OC increases.

In the revised manuscript, the original expression (lines 294–296)
"Given that the relative difference in extraction efficiency of total OC between MeOH and DMF was less than 10%, low-volatile OC should contain stronger light-absorbing chromophores (Saleh et al., 2014)." was changed to

"Given that the relative difference in extraction efficiency of total OC between MeOH and DMF was less than 10% and DMF dissolved more OC3 and OC4 than other solvents (Table 1), low-volatility OC should contain stronger light-absorbing chromophores (Saleh et al., 2014) and its mass fraction might determine the difference in BrC absorption across solvent extraction methods." (Lines 333–337)

**7.** Page 13, lines 298–299. "This is because the light absorption of DMF extracts depends less on wavelengths than other solvent extracts ($\mathring{A}$ ~4.5, Table 2)."
Page 14, lines 339–341. "In comparison to $\mathring{A}_m$ (6.81± 1.64; Table 3), the lower average $\mathring{A}_d$ (5.25 ± 0.64, $p < 0.01$) supports that more-absorbing BrC had less spectral dependence than less-absorbing BrC."
In Tables 2 and 3, there seems to be a negative relationship between the MAE and $\mathring{A}$ values. To illustrate that strong BrC chromophores had less spectral dependence than weak ones, I would suggest showing the relationship visually by plotting MAE vs. $\mathring{A}$.

*Reply:*

The linear relationships between average $\mathring{A}$ and $MAE_{365/550}$ of individual solvent extracts in Table 2 are provided in Figure S5, and those for $MAE_{365,\,d}$ versus $\mathring{A}_d$ and $MAE_{365,\,m}$ versus $\mathring{A}_m$ are shown in Figure S7.

In the manuscript, the original expression
"This is because the light absorption of DMF extracts depends less on wavelengths than other solvent extracts ($\mathring{A}$ ~4.5, Table 2)." (lines 298–299) was changed to

"This is because the light absorption of DMF extracts that contain stronger BrC chromophores depends less on wavelengths than other solvent extracts ($\mathring{A}$ ~4.5, Table 2). As shown in Figure S5, average $\mathring{A}$ and $MAE_{365/550}$ values of individual solvent extracts in Table 2 are negatively correlated." (Lines 339–342)

And the original expression
"In comparison to $\mathring{A}_m$ (6.81± 1.64; Table 3), the lower average $\mathring{A}_d$ (5.25 ± 0.64, $p < 0.01$) supports that more-absorbing BrC had less spectral dependence than less-absorbing BrC." (lines 339–341) was changed to

"Both $MAE_{365,d}$ and $MAE_{365,m}$ were negatively correlated ($p < 0.01$) with their corresponding $\mathring{A}$ values (Figure S7), and the lower average $\mathring{A}_d$ (5.25 ± 0.64, $p < 0.01$)

compared to $\text{Å}_m$ (6.81± 1.64; Table 3) supports that more-absorbing BrC had less spectral dependence than less-absorbing BrC." (Lines 400–403)

8. Figures 2 and 3. I would suggest the authors to put Abs365, MAE365, and Å on the y-axis.

*Reply:*

   Figures 2 and 3 have been revised as suggested.

**References**

Saleh, R., Robinson, E. S., Tkacik, D. S., Ahern, A. T., Liu, S., Aiken, A. C., Sullivan, R. C., Presto, A. A., Dubey, M. K., Yokelson, R. J., Donahue, N. M., and Robinson, A. L.: Brownness of organics in aerosols from biomass burning linked to their black carbon content, Nat. Geosci., 7, 647-650, https://doi.org/10.1038/ngeo2220, 2014.

---

## Author Comment (AC2)

This study compares the extraction of ambient PM$_{2.5}$ samples applying different solvents and the subsequent light absorption and determination of brown carbon (BrC). Authors find that the traditional approaches using MeOH or water extraction underestimate BrC absorption due to the insolubility of OC possessing larger chromophores and DMF exhibits the highest extraction efficiency among all the tested solvents. They suggest that using DMF instead of MeOH for BrC extraction and incorporate the results into receptor model will generate distinct source apportionment results. After PMF analysis, they conclude that the contributions of BrC from unburned fossil fuels and polymerization of aerosol organics are underestimated particularly. I do appreciate the interesting work and the information provides new insights into the radiative forcing of BrC. The work is well drafted, and I recommend publication in ACP before a few comments to be addressed as below.

**1.** Line 146-147. In the sampling setup, PUF is attached after two quartz filters to collect the gas phase polar and non-polar organic compounds. However, we do not see the subsequent treatment of the gas phase samples. Also, the absorption of vapors to quartz filter is substantial. In this regard, the sampling artifacts of this experimental design may be great concern and should be addressed.

*Reply:*

The adsorbent samples were analyzed for gas-phase organic molecular markers (OMMs), not BrC absorption. Details on the measurements and sampling artifacts of gas- and particle-phase OMMs have been provided in our previous studies (Gou et al., 2021; Qin et al., 2021). Similar to Xie et al. (2022), total concentrations (gas + particle phases) of OMMs were input for PMF modeling to avoid the influence of gas-particle partitioning. The total concentrations of individual OMMs were calculated as the sum of concentrations in Q$_f$, Q$_b$, and adsorbent samples, and were not impacted by the adsorption of organic vapors on quartz filters. In the original manuscript, we mentioned that

"The input bulk components and organic molecular marker (OMM) data for PMF analysis were obtained from Xie et al. (2022) and are summarized in Table S3." (Lines 231–233)

To make this clear, we added the following statements in the revised manuscript.
"The measurement results of gas- and particle-phase organic compounds were provided by Gou et al. (2021) and Qin et al. (2021)." (Lines 177–179)

"The total concentration data (Q$_f$ + Q$_b$ + adsorbent) of organic compounds have been used to apportion the light absorption of MeOH-soluble OC to specific sources (Xie et al., 2022), so as to avoid the impacts of gas-particle partitioning. In this work, the input particulate bulk components and total organic molecular marker (OMM) data for PMF analysis were obtained from Xie et al. (2022) and are summarized in Table S3." (Lines 265–270)

Additionally, the OC adsorbed on Q$_b$ and its light absorption were used to address positive sampling artifacts in *Section 2.3*. As shown in Table S6, the average Abs$_{365}$ of Q$_b$ samples is less than 10% of Q$_f$ samples. The light-absorbing properties of DMF extractable OC after Q$_b$ corrections are shown in Table 3 and Figure 2.

**2.** Session 3.3 PMF analysis. Current discussion about the PMF is brief, and the following key information should be included, either in the main text or the SI. (1) the stability test of the final solution, as it indicates the robustness of the solution. A solution fails the robustness test is meaningless. (2) The change of the $Q_{robust}/Q_{exp}$ with factor numbers should be examined.

*Reply:*

In comparison to the source apportionment performed by Xie et al. (2022), the input data set of this study only replaced the light absorption of water extracts with DMF extracts ($Abs_{365,d}$). Considering that the light absorption of aerosol extracts in water, MeOH, and DMF was intercorrelated ($r > 0.80$), these two studies are expected to have similar PMF error estimation results and $Q/Q_{exp}$ values. Xie et al. (2022) provided summaries of BS, DISP, BS-DISP error estimation diagnostics and $Q/Q_{exp}$ values for 4- to 10-factor PMF solutions as follows

|  | 4-factor | 5-factor | 6-factor | 7-factor | 8-factor | 9-factor | 10-factor |
|---|---|---|---|---|---|---|---|
| **BS diagnostics** | | | | | | | |
| Lowest %BS mapping | 76 | 64 | 44 | 27 | 40 | 27 | 30 |
| Highest % unmapped | 16 | 32 | 53 | 49 | 58 | 67 | 66 |
| | | | | | | | |
| **DISP diagnostics** | | | | | | | |
| Error Code: | 0 | 0 | 0 | 0 | 0 | 0 | 0 |
| Largest Decrease in Q: | 0 | -0.23 | -1.25 | -1.40 | -0.21 | -11.7 | -1.65 |
| %dQ: | 0 | -0.0015 | -0.010 | -0.013 | -0.0022 | -0.14 | -0.022 |
| Highest swaps by factor: | 0 | 0 | 0 | 0 | 0 | 0 | 0 |
| | | | | | | | |
| **BS-DISP Diagnostics** | | | | | | | |
| Number of cases accepted | 85 | 88 | 79 | 66 | 69 | 69 | 53 |
| % of cases accepted | 85% | 88% | 79% | 66% | 69% | 69% | 53% |
| Largest decrease in Q | -73.7 | -174 | -758 | 1996 | 1894 | -320 | 1182 |
| %dQ | -0.41 | -1.16 | -6.12 | 18.4 | 20.0 | -3.83 | 16.1 |
| Number of decreases in Q | 14 | 10 | 19 | 11 | 19 | 14 | 14 |
| Number of swaps in best fit | 0 | 1 | 0 | 11 | 4 | 5 | 13 |
| Number of swaps in DISP | 1 | 1 | 2 | 12 | 8 | 12 | 20 |
| Highest swaps by factor: | 0 | 2 | 0 | 10 | 3 | 8 | 11 |
| | | | | | | | |
| **Q/Qexp** | 3.52 | 2.94 | 2.49 | 2.18 | 1.97 | 1.78 | 1.63 |

In the revised supplementary information, we added a section (Text S2) describing the preparation of the input data set and the determination of the final factor number, including robustness analysis and $Q/Q_{exp}$ changes.

"*Text S2. PMF data preparation and factor number determination*
Similar to Xie et al. (2022), 102 observations of 9 $PM_{2.5}$ bulk components ($NH_4^+$, $SO_4^{2-}$, $NO_3^-$, $Ca^{2+}$, $Mg^{2+}$, OC, EC, WSOC and MEOC) and 50 OMMs (22 *n*-alkanes, 14 PAHs, 5 steranes and hopanes, C5-alkanetriols, 2-methyltetrols, levoglucosan, and 6 sugar and sugar alcohols) were selected to apportion the light absorption of aerosol extracts in methanol ($Abs_{365,m}$) and the solvent with the highest extraction efficiency ($\eta$) to sources. The measurement results of the bulk components in $PM_{2.5}$ and total OMMs

(gas + particle phase) are summarized in Table S3. Uncertainty fractions of bulk components and aerosol extract absorption were set to their ARPD values of collocated $Q_f$-$Q_b$ data (Yang et al., 2021; Xie et al., 2022; Figure 1). The uncertainties of OMM concentrations were calculated as (Zhang et al., 2009; Xie et al., 2016, 2019; Liu et al., 2017)

$$\text{Uncertainty} = \sqrt{(20\% \times \text{concentration})^2 + (0.5 \times \text{detection limit})^2} \qquad (3)$$

Missing values and measurements below detection limits (BDL) were replaced by the geometric mean of all observations and half of the detection limit, respectively. Their accompanying uncertainties were set to four times the geometric mean and five-sixths the detection limit (Polissar et al., 1998).

Because the identified sources for BrC absorption are essential, interpretability is the primary basis for determining an appropriate factor number and is defined by how PMF apportioned specific source-related OMMs (Shrivastava et al., 2007). Furthermore, the change in $Q/Q_{exp}$ with varying factor numbers is also a typical indicator of factor number selection (Liu et al., 2017; Wang et al., 2017, 2018). Specifically, $Q/Q_{exp}$ is expected to change less dramatically when the factor number increases to a certain value. The EPA PMF5.0 tool can evaluate the robustness of individual base-case solutions with three built-in error estimation methods, including bootstrapping (BS), displacement (DISP), and BS-DISP (Norris et al., 2014; Paatero et al., 2014; Brown et al., 2015). In this work, 100 BS runs were conducted with a minimum $r$ value of 0.8 (default 0.6) to map the BS run to base run factors. Once the error code or swap counts at dQmax=4 of DISP analysis were not 0, the base case solution was considered invalid. All input species were included for BS-DISP analysis.

In Table S4, $Q/Q_{exp}$ changes by 9.14% from 8- to 10-factor solutions, less significant than the value (10.0%–15.1%) for factor numbers varying from 4 to 8, indicating that a factor number of eight is needed to explain the input data. When examining the factor profiles, the 8-factor solution had the most interpretable factor profiles by identifying a lubricating oil combustion factor (Figure S6). The 9-factor solution resolved an unexplainable factor characterized by a mixture of anthropogenic and natural source markers (e.g., steranes, $Ca^{2+}$, and saccharides). In comparison to the input data set for PMF analysis in Xie et al. (2022), this work replaces the light absorption of water extracts with DMF extracts at 365 nm ($Abs_{365,d}$). The error estimation results of these two studies were similar. Although the factor matching rate of the BS runs decreased as the factor number increased, the BS matching rate of the 8-factor solution was larger than 50% when the default minimum $r$ value (0.6) was used. Furthermore, no DISP swap was observed and the acceptance rates of BS-DISP analysis were higher than 50% for 4- to 10-factor solutions. Therefore, the resulting base-case solutions are valid and interpretable, and an 8-factor solution was finalized to explain the sources of aerosol extract absorption."

We also mentioned this information in the main text of the revised manuscript.

Lines 272–273
"More information on input data preparation and the factor number determination are provided in supplementary information (Text S2 and Table S4)."

Lines 421–423

"A final factor number of eight was determined based on the interpretability of different base-case solutions (four to ten factors), the change in Q/Q$_{exp}$ with factor numbers, and robustness analysis (Text S2 and Table S4)."

**3.** Figure S5 UV-VIS spectra of 4-nitrophenol and 4-nitrocatechol. There is a strong light absorption at around 450 nm using DMF, which is not observed in other samples. It looks that unknown reactions occur, and the products introduce the unexpected light absorption. Considering that 4-nitrophenol and 4-nitrocatechol are representative tracers for biomass burning, readers may concern that DMF extracts would cause significant bias when investigate the BB BrC.

*Reply:*

Referring to existing studies, 4-nitrophenol and 4-nitrocatechol are strong light-absorbing chromophores coming from several sources, including biomass/biofuel burning (Lin et al., 2016, 2017; Xie et al., 2019), fossil fuel combustion (Lu et al., 2019), and photochemical reactions of aromatic VOCs with NO$_X$ (Xie et al., 2017). Therefore, these two species are not uniquely linked with biomass burning.

The strong light absorption of 4-nitrophenol and 4-nitrocatechol in DMF at 450 nm was not observed in other solvents, and was likely caused by unknown reactions. Then the solvent effect introduced by DMF might overestimate the light absorption of low-molecular-weight (LMW) nitrophenol-like species at > 400 nm in source or ambient aerosols. However, evidence shows that BrC absorption is dominated by large molecules with extremely low volatility (Saleh et al., 2014; Di Lorenzo and Young, 2016; Di Lorenzo et al., 2017), and LMW nitrophenol-like species have small contributions to particulate OM (e.g., < 1%) and aerosol extract absorption (e.g., <10%) (Xie et al., 2019, 2020; Li et al., 2020). The shapes of the light absorption spectra of aerosol extracts in DMF were similar to other solvents (Figure S4) and PAH solutions (Figure S6g-l), and no elevation in light absorption appeared at 400–500 nm (Figure S4). Thus, the overestimated absorption of LMW nitrophenol-like species in DMF might not substantially impact the overall BrC absorption of aerosol extracts.

These discussions have been added to the revised manuscript.
"In Figure S6, the absorbance spectra of 4-nitrophenol and 4-nitrocatechol in water shift toward longer wavelengths compared to their MeOH solution. This is because neutral and deprotonated forms of 4-nitrophenol and 4-nitrocatechol may have different absorbance spectra, and these two compounds are deprotonated at pH ≈ 7 (Lin et al., 2015b, 2017). The strong light absorption of 4-nitrophenol and 4-nitrocatechol in DMF at 450 nm was not observed in other solvents, and was likely caused by unknown reactions. Then the solvent effect introduced by DMF might overestimate the light absorption of low-molecular-weight (LMW) nitrophenol-like species at > 400 nm in source or ambient aerosols. Evidence shows that BrC absorption is dominated by large molecules with extremely low volatility (Saleh et al., 2014; Di Lorenzo and Young, 2016; Di Lorenzo et al., 2017), and LMW nitrophenol-like species have very low contributions to particulate OM (e.g., < 1%) and aerosol extract absorption (e.g., <10%) (Mohr et al., 2013; Zhang et al., 2013; Teich et al., 2017; Xie et al., 2019a, 2020; Li et al., 2020). The shapes of the light absorption spectra of aerosol extracts in DMF were similar to other solvents (Figure S4) and PAH solutions (Figure S6g-l), and no elevation in light absorption appeared at 400–500 nm. Thus, the overestimated absorption of

LMW nitrophenol-like species in DMF might not substantially impact the overall BrC absorption of aerosol extracts." (Lines 353–370)

**4.** Line 317-318. The authors propose that the low-volatility OC fractions are possibly featured with PAH skeleton and DMF has higher dissolubility for those compounds than MeOH. Nevertheless, no light absorbance difference is observed in Figure S5 g-l.

*Reply:*

In this work, we showed the difference in light absorption of ambient aerosol extracts across five solvents. However, the difference might be partly ascribed to the solvent effect, as solutions of the same compound in different solvents might have different light absorbance spectra.

In section 2.1, we provided a method for solvent effect evaluation.

"The solvent effect is not uncommon when measuring aerosol extract absorbance in difference solvents (Chen and Bond, 2010; Mo et al., 2017; Moschos et al., 2021), but is rarely accounted for in previous studies. To evaluate the influence of solvent effects on light absorption of different solvent extracts of the same sample, solutions of 4-nitrophenol at 1.90 mg $L^{-1}$, 4-nitrocatechol at 1.84 mg $L^{-1}$, and 25-PAH mixtures (Table S2) at 0.0080 mg $L^{-1}$ and 0.024 mg $L^{-1}$ (each species) in the five solvents and solvent mixtures were made up for five times and analyzed for UV/Vis spectra. The absorbance of PAH mixtures in water was not provided due to their low solubility." (Lines 227–234)

Figure S6 shows that PAH solutions have very similar absorbance spectra across the five solvents, indicating that the solvent effect does not impact the light absorption of organic compounds with a PAH structure. According to the results shown in Tables 1 and 2, DMF extracts of ambient aerosols contain more low-volatility OC (OC3 and OC4) and have higher light absorption than other solvent extracts. Considering that low-volatility OC is less water soluble and has a high degree of conjugation probably featured by a PAH structure, the large difference in light absorption between DMF and other solvent extracts is likely caused by the fact that DMF can dissolve more low-volatility OC.

Therefore, Table S5 and Figure S6 are used to demonstrate that the solvent effect has little influence on the light absorption of PAHs in different solvents. We did not identify the low-volatility PAHs only soluble in DMF in this work.

**5.** What are the 25 PAHs in the mixture solution and can you give some example structures that DMF have higher solubility than MeOH.

*Reply:*

The species information of the 25-PAH mixture is provided in Table S2 of the supplementary information. As we replied to the reviewer's 4[th] comment, the PAH mixture was used to evaluate the solvent impact on light absorption, and all the 25 species were dissolved.

Since we did not perform organic speciation for DMF and MEOH extracts, the structure that DMF has higher solubility than MeOH was not identified.

In the revised manuscript, we added

"However, we cannot rule out the impact of solvent effects on the comparison of light absorption spectra between MeOH and DMF extracts (Figure S4), and more work is warranted in identifying the structures more soluble in DMF than in MeOH." (Lines 378–381)

**6.** Line 283-284. As the author put it, the lower capability of MeOH in dissolving low-volatility OC fractions (OC3 and OC4) would lead to an underestimation of BrC absorption. Can you give an estimation of the underestimation so that the readers have intuitive knowledge?

*Reply:*

Based on the light absorption measurements of collocated samples from 09/2018–09/2019 in suburban Nanjing, the average $Abs_{365,d}$ was 30.7% higher than $Abs_{365,m}$ after $Q_b$ corrections. But the underestimation might vary with the time and location due to the changes in BrC sources.

The difference in $Abs_{365}$ between DMF and MeOH extracts were provided in the original manuscript.

"As shown in Table 3, average $Abs_{365,d}$ and $MAE_{365,d}$ values were 30.7% ($p < 0.01$) and 17.3% ($p < 0.05$) larger than average $Abs_{365,m}$ and $MAE_{365,m}$." (Lines 334–336)

In the revised manuscript, we added some text in the abstract and conclusions to show the difference and limits.

[revised manuscript text omitted]

---

## Author Comment (AC3)

In this manuscript, the authors investigate the light absorption characteristics of solvent-extractable brown carbon aerosol in ambient samples affected by multiple sources. The manuscript raises the topic of the effect of the selected solvent, or solvent mixture, on the extraction efficiency of organics and subsequently on the absorbance measured by UV-visible spectroscopy and associated light absorption properties (mass-normalized absorbance and its wavelength dependence). They find that N, N-dimethylformamide dissolves BrC associated with unburned fossil fuel and polymerization processes of aerosol organics more efficiently than methanol. The study results and potential implications are of interest to readers of *Atmospheric Chemistry and Physics*. Nevertheless, I would only recommend publication of this manuscript upon careful consideration by the authors of the specific comments below and subsequent clarifications and fine-tuning of some discussions in the revised manuscript.

**Specific comments**
**1.** Line 1: the message of the title may not be universal and could be considered misleading. If the authors prefer the current title to largely remain (although Reviewer 1 has already provided an alternative that may better reflect the paper content), I strongly recommend they change it to: "Potential underestimation…". The main reason is that methanol still efficiently extracted biomass burning-related BrC, which has been more widely studied in the literature.

*Reply:*

Thanks for the reviewer's suggestion. In this work, only ambient OC was extracted using different solvents. The results showed that the MeOH extraction method had a lower extraction efficiency than the DMF extraction method, and underestimated ambient BrC absorption. But methanol can still extract biomass burning BrC efficiently (> 90%).
In the revised manuscript, we changed the title to
"*Potential* underestimation of *ambient* brown carbon absorption based on the methanol extraction method and its impacts on source analysis"

**2.** Lines 44-45: for completeness, please clarify the following details in the abstract: (1) BrC aerosols associated with biomass burning, and coal (?), combustion sources were still highly soluble in MeOH; (2) the different MeOH solubility of BrC from different (seasonal) sources was likely the main reason for the aforementioned distinct time series. Therefore, (3) a more accurate alternative to the sentence: "These results highlight the necessity of replacing MeOH with DMF for further investigations on structures…" could be: "These results highlight the importance of testing different solvents to investigate the structures and light absorption of BrC, particularly of the low-volatility fraction potentially associated with certain non-traditional sources.". Please also rephrase related statements in Lines 103-104, 105-107, 227-228 ("potential underestimation"), 415, 418 ("may sometimes"), 423-424 ("…may potentially underestimate the contribution of solvent-extractable…"), and elsewhere if applicable.

*Reply:*

The original expression in lines 42–46 was changed to
"Source apportionment results indicated that the MeOH solubility of BrC associated with biomass burning, lubricating oil combustion, and coal combustion is

similar to their DMF solubility. The BrC linked with unburned fossil fuels and polymerization processes of aerosol organics was less soluble in MeOH than in DMF, which was likely the main reason for the large difference in time series between MeOH and DMF extract absorption. These results highlight the importance of testing different solvents to investigate the structures and light absorption of BrC, particularly for the low-volatility fraction potentially originating from non-combustion sources." (Lines 46–54)

The original expressions in lines 103–107 and 227-228 were changed to
"By comparing with the study results in Xie et al. (2022), this study evaluated potential underestimation of BrC absorption in methanol and its impacts on BrC source attributions. These results suggest that different solvents should be used in future investigations on the absorption, composition, sources, and formation pathways of low-volatility BrC." (Lines 132–136)

"To examine the influence of potential BrC underestimation based on the methanol extraction method on source apportionment, ………" (Lines 261–262)

and the original expressions in lines 415, 418, and 423–424 were changed to
"Comparisons of extraction efficiencies and light absorption of ambient aerosol extracts across selected solvents and solvent mixtures indicate that MeOH may sometimes be replaced with DMF for measuring BrC absorption, as low-volatility OC fractions containing strong chromophores are less soluble in MeOH than in DMF." (Lines 490–493)

"……, and had a potential to underestimate the contribution of BrC to total aerosol absorption." (Lines 497–498)

**3.** Line 93: the sentence is more informative with the following (or similar) addition referring to the extractable aerosol fraction: "…directly if the latter is not converted to particulate absorption with Mie calculations, solvent-matrix, and pH effects are not accounted for, and solvent solubility is not high.".

*Reply:*

Thanks for the reviewer's suggestion, and the original expression was changed to
"Given that the solvent extract absorption is not converted to particulate absorption with Mie calculations, solvent and pH effects are not accounted for, and BrC is not completely dissolved in typical solvents (e.g., water and methanol), BrC absorption in particles and solution can hardly be compared directly." (Lines 118–122)

**4.** Line 120: that is likely true for DMF; THF has been tested for biomass burning-influenced ambient BrC (Moschos et al., 2021); do the two observations agree? The authors state "rarely" in this sentence: do any other studies exist that have tested any of these two solvents for extracting BrC aerosol?

*Reply:*

Moschos et al. (2021) selected methanol to extract ambient aerosols based on the comparison of the absorbance with five other solvents: water, acetonitrile, acetone, tetrahydrofuran (THF), and dichloromethane. Only a winter and summer filters were extracted using different solvents, and the light absorption of THF extracts was much lower than that of MeOH extracts, which was consistent with the results in this study.

To the best of our knowledge, no other study has ever tested THF and DMF for extracting BrC.

The original expression was changed to

"Except for water and MeOH, DCM and THF were rarely used to extract OC for light absorption measurements (Cheng et al., 2021; Moschos et al. 2021), and DMF has not ever been tested for extracting BrC in literature." (Lines 149–151)

**5.** Line 195: when measuring the absorbance of solvent extracts, solvent-matrix effects (Reichardt, 2003) are not uncommon (yet rarely accounted for in the BrC research). Chen and Bond (2010; cited in the preprint), Mo et al. (2017), and Moschos et al. (2021) observed higher absorbance of water-extracted BrC aerosol that was further diluted/re-dissolved in methanol (for the same total extract volume). Could the authors discuss, in the revised manuscript, similar effects for their selected solvents/mixtures, as well as the implications for the results presented here when not correcting for such (i.e., currently, the solvent-matrix vs. solubility effects are not decoupled)? Can the authors rule out a solvent-matrix effect that would affect the wavelength-dependent comparison between MeOH and DMF, for example, in Fig. S4?

*Reply:*

As we replied to reviewer 2's 4th comment, the difference in light absorption of aerosol extracts across solvents might be partly ascribed to the solvent effect, as the same compound in different solvents might have different light absorbance spectra. To evaluate the influence of solvent effects on aerosol extract absorption, the light absorbance of typical BrC chromophores (4-nitrophenol, 4-nitrocatechol, and PAHs) in different solvents were compared (Lines 227–234; Table S5 and Figure S6).

Although the difference in light absorption between MeOH and DMF extracts is likely attributed to the fact that the low-volatility OC in ambient aerosols is more soluble in DMF, we cannot confirm that the solvent effect has no influence.

In the revised manuscript, we added more discussions on the potential influences of solvent effects on aerosol extract absorption.

"In Figure S6, the absorbance spectra of 4-nitrophenol and 4-nitrocatechol in water shift toward longer wavelengths compared to their MeOH solution. This is because neutral and deprotonated forms of 4-nitrophenol and 4-nitrocatechol may have different absorbance spectra, and these two compounds are deprotonated at pH ≈ 7 (Lin et al., 2015b, 2017). The strong light absorption of 4-nitrophenol and 4-nitrocatechol in DMF at 450 nm was not observed in other solvents, and was likely caused by unknown reactions. Then the solvent effect introduced by DMF might overestimate the light absorption of low-molecular-weight (LMW) nitrophenol-like species at > 400 nm in source or ambient aerosols. Evidence shows that BrC absorption is dominated by large molecules with extremely low volatility (Saleh et al., 2014; Di Lorenzo and Young, 2016; Di Lorenzo et al., 2017), and LMW nitrophenol-like species have very low

contributions to particulate OM (e.g., < 1%) and aerosol extract absorption (e.g., <10%) (Mohr et al., 2013; Zhang et al., 2013; Teich et al., 2017; Xie et al., 2019a, 2020; Li et al., 2020). The shapes of the light absorption spectra of aerosol extracts in DMF were similar to other solvents (Figure S4) and PAH solutions (Figure S6g-l), and no elevation in light absorption appeared at 400–500 nm. Thus, the overestimated absorption of LMW nitrophenol-like species in DMF might not substantially impact the overall BrC absorption of aerosol extracts. Furthermore, the absorbance of 4-nitrophenol and 4-nitrocatechol in DMF at 365 nm ($A_{365}$) was lower than that in MeOH, and PAH solutions showed very similar absorbance spectra across the five solvents (Figure S6g–l and Table S5). Considering that low-volatility OC fractions (e.g., OC3 and OC4) in the ambient are less water soluble (Table 1) and have a high degree of conjugation (Chen and Bond, 2010; Lin et al., 2014), their structures are probably featured by a PAH skeleton. Therefore, the large difference in $Abs_{365}$ between DMF and MeOH extracts (Table 2) was primarily ascribed to the fact that DMF can dissolve more OC3 and OC4 than methanol (Table 1). However, we cannot rule out the impact of solvent effects on the comparison of light absorption spectra between MeOH and DMF extracts (Figure S4), and more work is warranted in identifying the structures more soluble in DMF than in MeOH." (Lines 353–381)

Two statements were added to the method and conclusion sections.

"The solvent effect is not uncommon when measuring aerosol extract absorbance in difference solvents (Chen and Bond, 2010; Mo et al., 2017; Moschos et al., 2021), but is rarely accounted for in previous studies." (Lines 227–229)

"However, the influence of the solvent effect was not accounted for in this work when comparing the light absorption of different solvent extracts." (Lines 499–500)

**6.** Line 312: here, the authors have the opportunity to discuss also potential pH effects, e.g., the absorbance red-shift for the water-extract of 4-nitro-catechol in Fig. S5 and the isosbestic point ~365 nm, which seem to be consistent with the observation of Lin et al. (2017) for water vs. organic-solvent BrC aerosol extracts.

*Reply:*

Here we added the following discussions in the revised manuscript
"In Figure S6, the absorbance spectra of 4-nitrophenol and 4-nitrocatechol in water shift toward longer wavelengths compared to their MeOH solution. This is because neutral and deprotonated forms of 4-nitrophenol and 4-nitrocatechol may have different absorbance spectra, and these two compounds are deprotonated at pH ≈ 7 (Lin et al., 2015b, 2017)." (Lines 353–357)

**7.** Line 426: please clarify: "…in cold periods, when coal/biomass burning sources dominated the aerosol emissions..." if that is the case.

*Reply:*

Yes, that's true. The original expression was changed to

"Although light-absorbing properties of DMF and MeOH extracts had good agreement in cold periods, when biomass and coal burning sources dominated BrC emissions, their distinct time series in spring and summer implies that the contributions of certain BrC sources were underestimated or missed when the MeOH extraction method was used." (Lines 506–510)

**8.** Conclusions and implications section: It is important here to provide a broader view that will allow future studies to confirm these observations, while other approaches may still be helpful: for example, the authors could state that a combination of solvents with a broad polarity index (e.g., Lin et al., 2018) may still be good choice to cover different conditions, e.g., a mixture of non-polar (e.g., hexane), polar protic (e.g., MeOH) and polar aprotic solvents (e.g., DMF) for a range of BrC-containing samples influenced by different sources. Based on this and other comments above, there is no "universal evidence" from this study that DMF is the unique-best solvent for BrC under all conditions. At the same time, a more balanced discussion in the revised manuscript would encourage future studies to test DMF and potentially verify or revise the authors' observations.

*Reply:*

Thanks for the reviewer's suggestion. The first paragraph of the conclusions and implications section was changed to

"Comparisons of extraction efficiencies and light absorption of ambient aerosol extracts across selected solvents and solvent mixtures indicate that MeOH may sometimes be replaced with DMF for measuring BrC absorption, as low-volatility OC fractions containing strong chromophores are less soluble in MeOH than in DMF. Existing modeling studies on the radiative forcing of BrC (Feng et al., 2013; Wang et al., 2014; Zhang et al., 2020) often retrieved or estimated its optical properties from laboratory or ambient measurements based on water/methanol extraction methods (Chen and Bond, 2010; Hecobian et al., 2010; Liu et al., 2013; Zhang et al., 2013), and had a potential to underestimate the contribution of BrC to total aerosol absorption. However, the influence of the solvent effect was not accounted for in this work when comparing the light absorption of different solvent extracts. The difference between MeOH and DMF extract absorption might change with the time and location due to the variations in BrC sources. The results of this work also imply the necessity of applying different solvents or combinations of solvents with broad polarity and dissolving capability to study BrC composition and absorption, particularly for low-volatility fractions." (Lines 490–505)

**9.** Line 434: does that refer to the isoprene oxidation factor in Fig. 3? That is an important finding; the figure can be cited once more in this section together with this statement.

*Reply:*

This refers to the factors influenced by the polymerization processes of organic components. For example, the dust resuspension and isoprene oxidation factors.
Here we cited Fig. 3 again in the statement.

"……, but the structures and light-absorbing properties of potential polymerization products in ambient aerosols (Figure 3e, g) are less understood and warrant further study." (Lines 515–517)

**10.** Figure 3: please mention (possibly in the caption) that the biogenic emission factor *Abs* is below the detection limit (if that is the case).

Further, I agree with Reviewer 2 that it is critical to provide evidence for the robustness of the PMF solution for a reader to assess the quality of the results and the validity of the associated conclusions.

Could the authors also discuss the yearly evolution of the *Abs* relative difference between the two solvents for each PMF factor in Fig. 3? The relative difference seems low for coal and biomass burning throughout the year; what are the time-series trend and day-to-day variability for the other factors (those where both solvents seem to dissolve a non-negligible fraction of their chemical constituents)?

Finally, based on the statement in Lines 297-299, could the authors reproduce Fig. 3 for *Abs* at a longer wavelength and compare the two PMF results?

*Reply:*

The contributions of individual factors to $Abs_{365,d}$ and $Abs_{365,m}$ were output from PMF modeling, not measurements. Thus, the factor contributions cannot be compared with the detection limit.

As we replied to reviewer 2's second comment, we added a section (Text S2) describing the preparation of input data set and the determination of the final factor number, including robustness analysis and $Q/Q_{exp}$ changes (Table S4). The robust analysis results showed that the final PMF solution was valid and interpretable.

"*Text S2. PMF data preparation and factor number determination*
Similar to Xie et al. (2022), 102 observations of 9 $PM_{2.5}$ bulk components ($NH_4^+$, $SO_4^{2-}$, $NO_3^-$, $Ca^{2+}$, $Mg^{2+}$, OC, EC, WSOC and MEOC) and 50 OMMs (22 *n*-alkanes, 14 PAHs, 5 steranes and hopanes, C5-alkanetriols, 2-methyltetrols, levoglucosan, and 6 sugar and sugar alcohols) were selected to apportion the light absorption of aerosol extracts in methanol ($Abs_{365,m}$) and the solvent with the highest extraction efficiency ($\eta$) to sources. The measurement results of the bulk components in $PM_{2.5}$ and total OMMs (gas + particle phase) are summarized in Table S3. Uncertainty fractions of bulk components and aerosol extract absorption were set to their ARPD values of collocated $Q_f$-$Q_b$ data (Yang et al., 2021; Xie et al., 2022; Figure 1). The uncertainties of OMM concentrations were calculated as (Zhang et al., 2009; Xie et al., 2016, 2019; Liu et al., 2017)

$$\text{Uncertainty}=\sqrt{(20\%\times \text{concentration})^2+(0.5\times\text{detection limit})^2} \qquad (3)$$

Missing values and measurements below detection limits (BDL) were replaced by the geometric mean of all observations and half of the detection limit, respectively. Their accompanying uncertainties were set to four times the geometric mean and five-sixths the detection limit (Polissar et al., 1998).
Because the identified sources for BrC absorption are essential, interpretability is the primary basis for determining an appropriate factor number and is defined by how PMF apportioned specific source-related OMMs (Shrivastava et al., 2007).

Furthermore, the change in $Q/Q_{exp}$ with varying factor numbers is also a typical indicator of factor number selection (Liu et al., 2017; Wang et al., 2017, 2018). Specifically, $Q/Q_{exp}$ is expected to change less dramatically when the factor number increases to a certain value. The EPA PMF5.0 tool can evaluate the robustness of individual base-case solutions with three built-in error estimation methods, including bootstrapping (BS), displacement (DISP), and BS-DISP (Norris et al., 2014; Paatero et al., 2014; Brown et al., 2015). In this work, 100 BS runs were conducted with a minimum $r$ value of 0.8 (default 0.6) to map the BS run to base run factors. Once the error code or swap counts at dQmax=4 of DISP analysis were not 0, the base case solution was considered invalid. All input species were included for BS-DISP analysis.

In Table S4, $Q/Q_{exp}$ changes by 9.14% from 8- to 10-factor solutions, less significant than the value (10.0%–15.1%) for factor numbers varying from 4 to 8, indicating that a factor number of eight is needed to explain the input data. When examining the factor profiles, the 8-factor solution had the most interpretable factor profiles by identifying a lubricating oil combustion factor (Figure S8). The 9-factor solution resolved an unexplainable factor characterized by a mixture of anthropogenic and natural source markers (e.g., steranes, $Ca^{2+}$, and saccharides). In comparison to the input data set for PMF analysis in Xie et al. (2022), this work replaces the light absorption of water extracts with DMF extracts at 365 nm ($Abs_{365,d}$). The error estimation results of these two studies were similar. Although the factor matching rate of the BS runs decreased as the factor number increased, the BS matching rate of the 8-factor solution was larger than 50% when the default minimum $r$ value (0.6) was used. Furthermore, no DISP swap was observed and the acceptance rates of BS-DISP analysis were higher than 50% for 4- to 10-factor solutions. Therefore, the resulting base-case solutions are valid and interpretable, and an 8-factor solution was finalized to explain the sources of aerosol extract absorption."

The relative difference in contributions of each factor to $Abs_{365,d}$ and $Abs_{365,m}$ is the same throughout the year due to the PMF calculation method. Thus, the time series of the absolute difference is shown instead in Figure S9. The ARPD and COD values between factor contributions to $Abs_{365,d}$ and $Abs_{365,m}$ are also added to each plot in Figure S9.

More discussions on the difference in factor contributions to $Abs_{365,d}$ and $Abs_{365,m}$ were added in the revised manuscript.

"Figure 3 compares the time series of factor contributions to $Abs_{365,d}$ and $Abs_{365,m}$. ARPD and COD values between factor contributions to $Abs_{365,d}$ and $Abs_{365,m}$ and the absolute difference are exhibited in Figure S9. $Abs_{365,d}$ and $Abs_{365,m}$ had comparable contributions from biomass burning, lubricating oil combustion, and coal combustion (Figure 3a, c, d). The small COD values of these three factors (0.0041–0.17) indicated no significant divergence. The biogenic emission and isoprene oxidation factors exhibited complete difference (ARPD = 200%, COD = 1; Figure S9f, g) as they had no contribution to $Abs_{365,m}$. Among the eight factors, the non-combustion fossil, dust resuspension, and isoprene oxidation factors had the largest median difference in factor contributions to $Abs_{365,d}$ and $Abs_{365,m}$ (0.63–0.67 $Mm^{-1}$) with substantial heterogeneity (COD > 0.20), followed by the secondary inorganics factor (0.20 $Mm^{-1}$, COD = 0.41). The temporal variations of the absolute difference shown in Figure S9 are identical to the contributions of individual factors to $Abs_{365,d}$ or $Abs_{365,m}$ (Figure 3)." (Lines 444–457)

"This might explain the elevated difference between $Abs_{365,d}$ and $Abs_{365,m}$ contributions of the isoprene oxidation factor in summer (Figure S9g)." (Lines 472–

473)

The light absorption of methanol extracts becomes very weak at $\lambda > 400$ nm (e.g., Figure S4), and more than 20% of methanol extract samples have no light absorption at $\lambda = 450$, 500, and 550 nm. Due to the limit in observation number ($N = 102$), we performed PMF source apportionment for $Abs_{\lambda,d}$ and $Abs_{\lambda,m}$ at $\lambda = 400$ nm using the same speciation data, and reproduced Figure 3 at $\lambda = 400$ nm.

Table 1. Comparisons of light-absorbing coefficients of ambient $PM_{2.5}$ extracts in DMF and MeOH at 365 nm and 400 nm.

| | DMF | | | MeOH[a] | | |
|---|---|---|---|---|---|---|
| | Median | Mean ± std | Range | Median | Mean ± std | Range |
| $Abs_{365}$, Mm-1 | 6.99 | 8.42 ± 5.40 | 1.14–30.8 | 5.59 | 6.43 ± 4.66 | 0.38–29.6 |
| $Abs_{400}$, Mm-1 | 4.39 | 5.44 ± 3.55 | 0.76–19.9 | 3.18 | 3.88 ± 2.96 | 0.21–10.7 |

[a] Data for MeOH extracts were obtained from Xie et al. (2022).

As shown in the figure below, the biomass burning, lubricating oil combustion, and coal combustion have comparable contributions to $Abs_{400,d}$ and $Abs_{400,m}$; the non-combustion fossil, dust resuspension, and isoprene oxidation factors lead the difference between $Abs_{400,d}$ and $Abs_{400,m}$ contributions. These results still indicate that large BrC molecules from unburned fossil fuels and potential polymerization processes are less soluble in MeOH than in DMF. Our discussions and conclusions remain the same even if the wavelength is shifted to 400 nm. Therefore, we did not compare the two PMF results in the revised manuscript.

[Figure]

Figure 1. Time series of factor contributions to Abs$_{400}$ of DMF and MeOH extracts of ambient PM$_{2.5}$ samples.

**Technical corrections**

**1.** Lines 281, 295, 315 & 416: "low-volatility".

*Reply:*

The "low-volatile" was changed to "low-volatility" throughout the manuscript.

**2.** Table 3: correct the superscript to "Mm-1".

*Reply:*

Thanks. It has been corrected as suggested.

**3.** Table S3: please provide the units of the tabulated data other than *Abs*365.

*Reply:*

Units were added for other data in Table S3.

**References**

Moschos, V., Gysel-Beer, M., Modini, R. L., Corbin, J. C., Massabò, D., Costa, C., Danelli, S. G., Vlachou, A., Daellenbach, K. R., Szidat, S., Prati, P., Prévôt, A. S. H., Baltensperger, U., and El Haddad, I.: Source-specific light absorption by carbonaceous components in the complex aerosol matrix from yearly filter-based measurements, Atmos. Chem. Phys., 21, 12809-12833, 10.5194/acp-21-12809-2021, 2021.

---

## Author Response (AR2)

**Comments to the author**:
One of the reviewers have a minor follow-up comment that they hope you will address:
Lines 118–122: It is not "a given" that the solvent extract absorbance is not converted to particulate absorption with Mie calculations; certain studies have attempted this conversion. What could be highlighted in this sentence is that even in this case, pH and solvent matrix effects, as well as the potential incomplete solubility of BrC in common solvents, should still be considered in order to be able to reliably compare BrC absorption measured directly in particles vs. that derived from solvent extracts (with Mie calculations). I suggest rephrasing to remove ambiguity.

*Reply:*

The original expression was changed to

[revised manuscript text omitted]